# Music listening and stress recovery in healthy individuals: A systematic review with meta-analysis of experimental studies

**Krisna Adiasto**[1]*, **Debby G. J. Beckers**[1], **Madelon L. M. van Hooff**[1], **Karin Roelofs**[1,2], **Sabine A. E. Geurts**[1]

**1** Behavioural Science Institute, Radboud University, Nijmege, Netherlands, **2** Donders Institute for Brain, Cognition, and Behaviour, Radboud University, Nijmege, Netherlands

* k.adiasto@bsi.ru.nl

**Data Availability Statement:** All materials relevant to our review, including: (a) the pre-registered study protocol; (b) an outline of the search strategy; (c) a list of screened articles with rationales for exclusion; (d) the meta-analysis data

## Abstract

Effective stress recovery is crucial to prevent the long-term consequences of stress exposure. Studies have suggested that listening to music may be beneficial for stress reduction. Thus, music listening stands to be a promising method to promote effective recovery from exposure to daily stressors. Despite this, empirical support for this opinion has been largely equivocal. As such, to clarify the current literature, we conducted a systematic review with meta-analysis of randomized, controlled experimental studies investigating the effects of music listening on stress recovery in healthy individuals. In fourteen experimental studies, participants ($N = 706$) were first exposed to an acute laboratory stressor, following which they were either exposed to music or a control condition. A random-effects meta-regression with robust variance estimation demonstrated a non-significant cumulative effect of music listening on stress recovery $g = 0.15$, 95% CI [-0.21, 0.52], $t(13) = 0.92$, $p = 0.374$. In healthy individuals, the effects of music listening on stress recovery seemed to vary depending on musical genre, who selects the music, musical tempo, and type of stress recovery outcome. However, considering the significant heterogeneity between the modest number of included studies, no definite conclusions may currently be drawn about the effects of music listening on the short-term stress recovery process of healthy individuals. Suggestions for future research are discussed.

## Introduction

The prevalence of stress-related diseases worldwide has seen no decrease over the previous decade [1, 2], as stress has become so pervasive in daily life that our physiological systems are under constant pressure to cope with various stressors [3]. Stress recovery has been introduced as a process which may mitigate the adverse consequences of frequent stress exposure [4, 5]: effective stress recovery on a daily basis may prevent the occurrence of blunted or exaggerated stress responses that over time develop into various physiological and psychological disorders, such as cardiovascular and cerebrovascular disease, hypertension, burnout, and depression [2, 5–8].

set with extracted data; and, (e) R code to replicate the analysis, are available on the Open Science Framework (https://osf.io/9pxhj/?view_only=0f2f28db4adf4a2492aa57e5e003cc9f).

**Funding:** The author(s) received no specific funding for this work.

**Competing interests:** The authors have declared that no competing interests exist.

Given the importance of effective stress recovery from exposure to daily stressors, research on potential means to promote stress recovery has experienced significant growth [5]. Various activities have been proposed that may lead to better stress recovery, one among them being music listening. Music listening may have a modulatory effect on the human stress response [9]. Furthermore, given that music is readily available through online streaming services, music listening stands to be a time- and cost-effective method to facilitate daily stress recovery. Indeed, a recent meta-analysis of 104 randomized controlled trials on the effects of music concluded that music-based interventions have a positive impact on both physiological ($d = .380$, 95% CI [0.30–0.47]) and psychological ($d = .545$, 95% CI [0.43–0.66]) stress-related outcomes [10]. However, a large proportion of studies included in this meta-analysis were conducted in medical or therapeutic settings, and the included music-based interventions encompassed not only music listening but also music therapy. Thus, a more specific review to determine whether music listening alone is beneficial for the recovery of healthy individuals outside medical and therapeutic settings seemed justified.

To expand on the above considerations: stressors in medical or therapeutic settings (e.g., treatment anxiety, pregnancy, and labor) and their subsequent stress recovery processes can be difficult to generalize to more daily settings [10–13]. Next, with regards to music-based interventions, music listening simply involves listening to a particular song, while music therapy is characterized by the presence of a therapeutic process and use of personal music experiences, and thus must be performed by a trained music therapist [14]. In practice, music therapy may not only involve music listening, but also music playing, composing, songwriting, and interaction with music [10, 14]. The effects of music therapy on stress appear to be more consistent compared to music listening [10, 15, 16]. Studies on music listening and stress recovery in healthy individuals are indeed equivocal: though music listening is considered beneficial for physiological stress recovery, several studies have reported no differences in heart rate, heart rate variability, respiration rate, blood pressure, or cortisol recovery between participants who listened to music and those who either sat in silence or listened to an auditory control [17–20]. Similarly, although music is notable for its anxiolytic effects, several studies have reported no significant differences in post-stressor anxiety between participants who listened to music and those who did not [3, 18, 21]. Taken together, it is currently difficult to draw definite conclusions about the effects of music listening on stress recovery in healthy individuals, particularly outside medical and therapeutic settings [15, 22].

Therefore, to expand on previous reviews, we opted to conduct a systematic review with meta-analysis on experimental studies in healthy individuals, focusing specifically on the role of music *listening* in stress recovery. In our review, we focus specifically on experimental studies, under the assumption that greater control over study variables would help reduce between-study heterogeneity. Furthermore, considering the crucial role of stress recovery in preventing the long-term consequences of stress exposure [5, 23], we believe the acute stress responses elicited by laboratory stressors would more closely approximate typical stress responses in daily life. The aim of our review was two-fold: through systematic review, we provide a comprehensive account of experimental studies examining the effect of music listening on stress recovery. Through meta-analysis, we assess the reliability of the effect of music listening on stress recovery, including the extent and impact of publication bias, and weigh-in on outstanding discussions within existing literature.

## The stress response

The stress response can be conceptualized as a compensatory reaction aimed at mitigating the potential consequences of a stressor [24, 25]. The stress response is best illustrated by the

archetypal 'fight-flight-freeze' reaction: in the presence of a stressor, the brain initiates an elegant synergy of neuroendocrine, physiological, and psychological processes that serve to mobilize energy resources and direct attention towards prominent stimuli, with the aim of promoting appropriate and rapid action [26, 27]. During a stress response, the autonomic nervous system (ANS) suppresses parasympathetic activity and promotes sympathetic exertion, resulting in marked increases in heart rate, respiration rate, systolic and diastolic blood pressure, and salivary secretion of the dietary enzyme, alpha-amylase [27–31]. These changes are mediated by neuropeptides (e.g., corticotropin-releasing factor) and catecholamines (e.g., norepinephrine, dopamine) [24, 25]. Simultaneous with ANS activity, the hypothalamic-pituitary-adrenocortical (HPA) axis begins a process which leads to a surge of cortisol production in the adrenal cortex [24, 25]. Cortisol acts as a regulator of the stress response, whose effects occur in a temporally specific manner due to variations in corticosteroid receptor affinity and distribution throughout the body [24, 26, 32]. Cortisol may require up to 45 minutes to reach peak concentration levels, during which it binds to high-affinity corticosteroid receptors [24]. This process enables rapid, non-genomic effects that sustain ANS-mediated changes for the duration of the stressor, while suppressing immune system function [32–34]. This suppression is visible through lower concentrations of immunoglobulins, such as salivary immunoglobulin-A (s-IgA) [35].

The physiological changes triggered by the ANS and HPA axis are supplemented by psychological changes that motivate adaptive behaviours required to cope with the stressor [25, 27]. For example, the unpleasant feeling one gets when experiencing anxiety and negative affect in response to a stressor is thought to prompt behaviours aimed at reducing these unpleasant states. Since psychological reactions to stressors are contingent on how individuals perceive, evaluate, and react to threats and challenges [36], self-reported measures of stress, anxiety, arousal, and emotion are common in psychological research on stress and its consequences [18, 37–39].

## Stress recovery

The stress response is considered adaptive when it is short-lived and immediately followed by a period of recovery following stressor cessation. In this period, ANS- and HPA-mediated changes that have occurred in response to a stressor revert to pre-stress baselines [24, 25, 27]. Therefore, stress recovery may be conceptualized as the process of unwinding that is opposite to the neuroendocrine, physiological, and psychological activation that occurs during the stress response [4, 5]. Following a stress response, ANS-mediated changes quickly revert to pre-stress levels within 30 to 60 minutes [26]. This manifests as a restoration of parasympathetic activity, marked by a deceleration of heart rate and respiration rate, lower systolic and diastolic blood pressure, and less activity of salivary alpha-amylase [4, 28–31]. This restoration of parasympathetic activity typically precedes any decline in cortisol. Instead, during the same window of time, cortisol levels will have just reached their peak, activating low-affinity corticosteroid receptors [40]. This process is thought to signal the termination of the stress response, as the binding of cortisol to low-affinity receptors inhibits further autonomic activation [24, 26]. As cortisol levels begin to decrease, slow, cortisol-mediated genomic changes are initiated, which directly oppose the rapid effects of catecholamines and the non-genomic effects of cortisol [24, 26]. Following a stressor, these genomic changes may take up to one hour to commence and may continue for several hours [24, 26].

At a psychological level, stress recovery is typically experienced as a reduction of unpleasant states, which is often reflected by lower ratings of self-reported stress, anxiety, and negative affect, along with higher ratings of relaxation and positive affect [5, 15, 18]. However, it is

worth noting that persistent, ruminative thoughts about a stressor may delay stress recovery by prolonging the physiological activation that occurs during the stress response [41–45]. Indeed, participants who reported higher rumination following a stress task demonstrated poorer heart rate, systolic blood pressure, diastolic blood pressure, and cortisol recovery compared to participants who did not [41, 42, 44, 46, 47].

## Music listening and stress recovery

Within the current literature, music listening has frequently been related to various neuroendocrine, physiological, and psychological changes that are considered beneficial for stress recovery [10, 11, 15, 22]. For example, music listening has been associated with lower heart rate [48–50], systolic blood pressure [21, 49, 51], skin conductance [17, 19, 52, 53], and cortisol [54, 55] compared to silence or an auditory control condition. Furthermore, music listening has been associated with higher parasympathetic activity [56] compared to silence [3, 37]. Together, these findings suggest that music listening may generate beneficial changes in ANS and HPA axis activity that should be conducive to the stress recovery process [27, 57, 58]. Furthermore, studies have demonstrated that listening to music may influence mood [59, 60]. Indeed, music listening has been associated with lower negative affect [37], higher positive affect [18, 61], and fewer self-reported depressive symptoms [37] compared to silence or an auditory control condition. Music listening has also been associated with lower subjective stress [53, 54], lower state anxiety [37, 48, 49], and higher perceived relaxation [17, 48, 62].

The exact mechanisms underlying the effects of music listening on stress recovery remain to be elucidated. Music-evoked positive emotions are thought to be particularly beneficial for stress recovery, as they may help undo the unfavourable changes wrought by negative emotions during stress, ultimately aiding the stress recovery process [63]. Alternatively, music-evoked emotions may promote a more robust, and thus more adaptive, stress response [61], which may be followed by an equally robust period of stress recovery. Next, it has been theorized that music may act as an anchor that draws attention away from post-stressor ruminative thoughts or negative affective states, thus preventing a lengthening of physiological activation, and facilitating a more regular stress recovery process [45, 64]. Finally, physiological rhythms in our body, such as respiration, cardiovascular activity, and electroencephalographic activity, may become fully or partially synchronized with rhythmical elements perceived in music [65–68]. This rhythmic entrainment process is thought to occur via a bottom-up process that originates in the brainstem: salient musical features, such as tempo, pitch, and loudness, are continuously tracked by the brainstem, generating similar changes in ANS activity over time [69, 70]. Indeed, studies have demonstrated that changes in a song's musical envelope, which represents how a song unfolds over time, are closely followed by proportional changes in blood pressure and skin conductance [52, 65]. Similarly, incremental changes in musical tempo, which represents the speed or pace of a song, were predictive of similar changes in heart rate, blood pressure, and respiration rate [71–73]. It is further hypothesized that the physiological changes resulting from rhythmic entrainment may evoke any number of associated emotions via proprioceptive feedback mechanisms [66, 69, 70]. Indeed, higher self-reported entrainment predicted increased positive affect, along with other self-reported emotional responses, such as transcendence, wonder, power, and tenderness [66].

## Which music works best?

There are several ongoing discussions about potential moderating effects in the relationship between music listening and stress recovery. We briefly describe these effects below, and later contribute to the discussion through moderator analyses.

**Classical music vs. other genres.** Classical music is considered the golden standard in many stress management efforts. Indeed, a copious amount of 'anti-stress' playlists often feature some selection of classical pieces. To discern which music best promotes stress recovery, studies have contrasted the effects of classical music with other musical genres, including rock [48], jazz and pop [21], and heavy metal [17]. We compare the effects of different musical genres in our moderator analysis.

**Instrumental vs. lyrical.** It is commonly believed that instrumental, as opposed to lyrical music, would better promote stress recovery. However, several studies have argued that lyrics may act as a stronger distractor compared to the sound of instruments. Thus, lyrical music may be more effective than instrumental music in preventing the prolonged physiological activation that may occur due to ruminative thoughts [17, 18, 74]. We compare the effects of instrumental and lyrical music in our moderator analysis.

**Self- vs. experimenter selected.** Studies on the effects of music often fail to consider the differential effects of self-selected (i.e., chosen by participants) and experimenter selected (i.e., chosen for participants by the experimenter) music [15]. It is hypothesized that allowing participants to select their own music may be more helpful to promote stress recovery due to a restoration of perceived control [15]. It has also been argued that individuals select music in service of personal self-regulatory goals [64, 75, 76], meaning that individuals know precisely which music to select for them to effectively recover from stress [38, 77]. Furthermore, previous studies have found that listening to self-selected music may help elicit stronger and more positive emotional responses regardless of a song's valence (positive or negative) and arousal (high or low), possibly due to increased preference and familiarity towards the self-selected music [78–80]. In theory, self-selected music should thus be more beneficial compared to experimenter-selected music for the purpose of stress recovery. We compare the effects of self- and experimenter selected music in our moderator analysis.

**Fast vs. slow tempo.** Several studies have investigated whether listening to music with slower tempo will better facilitate stress recovery compared to music with faster tempo. For example, while listening to an instrumental song, proportional increases and decreases in tempo resulted in similar changes in participants' heart rate [73]. Similarly, sequential decreases in tempo predicted greater increases in parasympathetic activity compared to sequential increases in tempo [71]. We investigate whether slower tempi differentially influence the effect of music listening on stress recovery in our moderator analysis.

## Method

The present review was designed following the Preferred Reporting Items for Systematic Review and Meta-Analysis Protocols (PRISMA-P) guidelines [81]. All materials relevant to this review, including: (a) the pre-registered study protocol; (b) an outline of the search strategy; (c) a list of screened articles with rationales for exclusion; (d) the meta-analysis data set with extracted data; and, (e) R code to replicate the analysis reported in this review, are available on the Open Science Framework (https://osf.io/9pxhj).

### Study selection

The study selection process is summarized in Fig 1. In April 2021, we conducted a comprehensive literature search for experimental studies on the effect of music listening on stress recovery. The search was conducted using RUQuest, the electronic search system of Radboud University library, which accesses several notable bibliographic databases, including MEDLINE, Wiley Online Library, ScienceDirect, SpringerLink, Taylor and Francis, and JSTOR. The results of this primary search were supplemented with three additional electronic searches

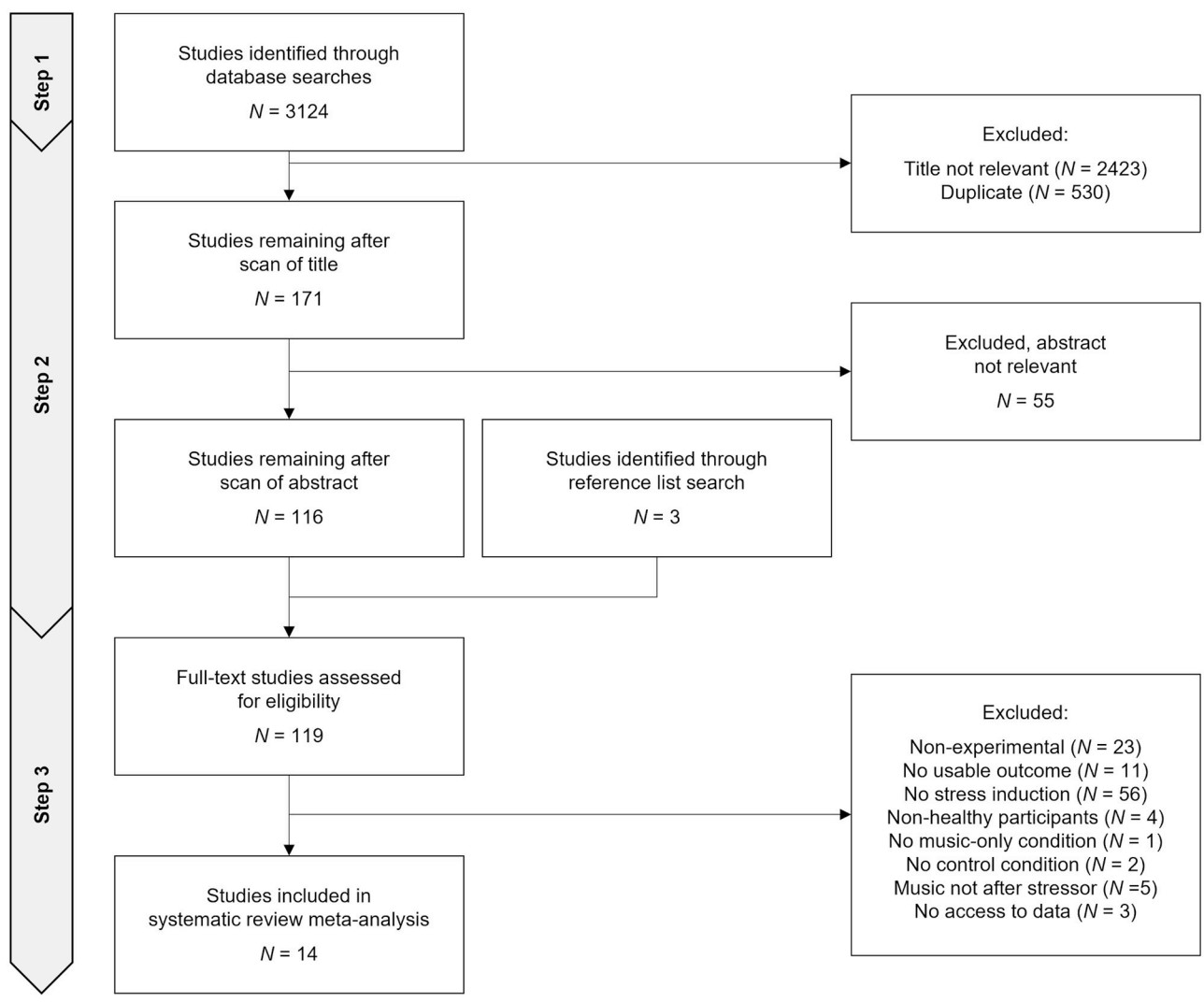

**Fig 1. Overview of the study selection process.**

in the publication databases of Web of Science, PsycINFO, and PubMed. Appendix A provides a complete description of our search terms. Together, this first step resulted in 3124 articles.

Next, the first author (KA) screened all titles and abstracts for studies examining the effects of music listening on stress recovery. If there was any doubt about the eligibility of an article, it was retained for further review. During this initial screening, 3008 articles were excluded. KA then scanned the reference lists of the 116 remaining articles for potentially relevant studies, resulting in an additional three articles. Together, this second step resulted in 119 full-text reports to be assessed for eligibility.

Lastly, KA used the following criteria to assess full-text reports for eligibility:

First, to minimize between-study heterogeneity, and to ensure that included studies investigated the effects of music listening on stress recovery as precisely as possible, studies must employ an experimental design including stress induction, with random assignment of participants to experimental and control conditions. Quasi-experimental studies were included only when they incorporated a control or comparison group. Second, studies should compare music listening to silence or an auditory stimulus (e.g., white noise, audiobooks). To ensure

that included studies tested the immediate effect of music listening on stress recovery, exposure to music, silence, or auditory stimuli must occur after the stress induction procedure. Third, to demonstrate this effect, studies must include at least one measure of neuroendocrine (e.g., cortisol), physiological (e.g., heart rate, blood pressure), or psychological (e.g., subjective stress, positive and negative affect) stress recovery outcome. Fourth, given that stress reactivity and recovery responses differ between children and adults, and with consideration to the potential role of music in the prevention of stress-related diseases in adults, studies must include healthy, adult, human participants. Fifth, to improve the generalization of our results in the context of daily stress recovery, studies where stress recovery occurred within a medical or therapeutic context, such as a hospital or operating room, were excluded. Finally, for the purpose of the meta-analysis, means and standard deviations of stress recovery outcomes following stressor cessation must be available. Corresponding authors were contacted when this information was not available. When authors did not or could not provide the required information (e.g., due to data no longer being accessible), outcomes were dropped from the meta-analysis. Following attempts to obtain missing information, the final sample for our review consisted of 14 studies.

## Methodological moderators of interest

Several methodological differences were identified between included studies that may moderate the effect of music listening on stress recovery:

**Stress induction procedures.** Studies utilized a diverse array of stress induction procedures. These include mental arithmetic tasks [e.g., 21], adaptations of the Trier Social Stress Task [e.g., 3], impromptu presentations [49, e.g., 50], unpleasant stimuli [e.g., 82], cognitive tests [e.g., 48], or a $CO_2$ stress task [61]. Stress induction procedures may generally be classified based on the inclusion of a social-evaluative threat (SET) component, which are designed to induce psychosocial stress and have been shown to elicit greater cardiovascular and cortisol responses [83]. In the event of a greater stress response, the effects of music listening on stress recovery may be larger, since there may be a larger window for the stress recovery process to occur. We examined this possibility in our moderator analysis.

**Stress induction checks.** Stress induction procedures in included studies were not always successful. Given that successful stress induction procedures are crucial to ensure that participants experience some physiological or psychological change they may recover from, in our moderator analysis we examined whether the effect of music listening on stress recovery differed based on the outcome of a study's stress induction check (manipulation check).

**Type of outcome.** Studies adopted numerous outcome measures as indicators of stress recovery. These include indicators related to ANS and HPA axis activity, such as heart rate [e.g., 49], heart rate variability [e.g., 3], blood pressure [e.g., 84], respiration rate [e.g., 17], skin conductance [58], salivary cortisol [e.g., 54], salivary alpha-amylase (sAA) [e.g., 38], and salivary immunoglobulin-A (sIgA) [e.g., 85], as well as indicators for psychological consequences of the stress response, such as subjective stress [e.g., 18], perceived relaxation [e.g., 17], state anxiety [e.g., 21], rumination [e.g., 18], and affect [e.g., 37]. In our moderator analysis, we examined whether the effects of music listening on stress recovery differed across general (neuroendocrine, physiological, psychological) and specific outcome types.

**Duration of music.** Studies differed with regards to how long participants listened to music following stressor cessation. This duration ranged from two minutes [e.g., 53] to forty-five minutes [e.g., 54]. We examined whether the effect of music listening on stress recovery differed based on duration of music listening.

## Data extraction, moderator coding, and quality assessment

KA extracted means, standard deviations, and total participants per condition for each stress recovery outcome. When these statistics were not included in text, but informative graphs were provided, KA used an open-source program to extract data from the graphs [86]. Coding criteria for each moderator can be found in Table 1. The '141–160 bpm', 'unsuccessful', 'salivary IgA', and 'salivary alpha-amylase' moderator levels were ultimately not included in the meta-analysis due to unavailable information.

Next, KA assessed the quality of included studies using the revised Cochrane risk of bias tool for randomized trials (RoB 2) [87]. Based on criteria in the RoB 2, studies with low risk of bias were considered high quality, while those with some concerns and high risk of bias were considered moderate and low quality respectively. Fig 2 summarizes the results of the quality assessment procedure.

Based on the RoB 2, all included studies were of moderate quality due to unavailable information on pre-specification of analysis plans. Thus, it was difficult to completely rule out bias that may have occurred due to a selection of reported results. Since the quality of included studies was homogenous, study quality was thus not included in our moderator analysis. An exploratory analysis with less stringent criteria, where potential risk of bias from selection of reported results is not included in our quality assessment procedure, is reported in Appendix B.

Data extraction, moderator coding, and quality assessment were conducted by KA in coordination with DB and MvH. Disagreements were resolved through face-to-face discussions, or through consultation with SG and KR when no consensus could be reached.

## Meta-analytic approach

**Effect size index.** We calculated Hedges' $g$ for each comparison using the escalc function of the metafor package [88] in R 3.6.3 [89]. In the present study, a Hedges' $g$ of zero indicates the effect of music listening on stress recovery is equivalent to silence or an auditory control. Conversely, a Hedges' $g$ greater than zero indicates the degree to which music listening is more effective than control, while a $g$ less than zero indicates the degree to which music listening is less effective than control. The effect sizes are reported in Table 2.

**Meta-analytic approach.** Due to use of multiple stress recovery outcomes, eleven out of fourteen studies included in the meta-analysis contributed multiple effect sizes of interest. To deal with the statistical dependency caused by the inclusion of multiple effect sizes from the same study, we use a combination of multivariate meta-regression [90] and robust variance estimation (RVE) [91] to estimate overall effect sizes and conduct moderator analyses. Although we believe our approach using RVE was the most suitable for our data, we also calculated overall effect sizes using the aggregation method outlined in Borenstein et al. [92], and random-effects meta-analyses without correcting for dependencies. These yielded estimates that were nearly identical to those generated by our approach and were therefore not reported.

**Outlier detection.** Currently, methods to identify outliers in meta-regression models with RVE are not yet available. Therefore, we first fit a random-effects meta-regression model without correcting for dependencies between effect sizes. Values for influential case diagnostics (e.g., covariance ratios, Cook's distance, studentized residuals) were subsequently requested using the 'influence' function of the 'metafor' package [88]. As this approach does not fully consider the nature of dependencies between effect sizes from each study, the results of this analysis were treated as a sensitivity analysis for the estimated overall effect of music listening on stress recovery. All extracted effect sizes were retained in further analyses.

**Table 1. Moderator coding criteria.**

| Moderator (bolded) and level | Criteria |
|---|---|
| **Classical vs. other genres** | |
| Classical | If no in-text description of genre was provided, the first author attempted to infer musical genre after listening to the reported musical stimuli. When this was also not possible, musical genre was coded as 'Unspecified'. |
| Heavy metal | |
| Jazz | |
| Pop | |
| Unspecified | |
| **Instrumental vs. lyrical** | |
| Instrumental | Music stimuli did not contain lyrics. |
| Lyrical | Music stimuli contained lyrics. |
| **Self- vs. experimenter selected** | |
| Self | Music stimuli selected by participants. |
| Experimenter | Music stimuli selected by the experimenter(s). |
| Pseudo | Music stimuli selected by participants from an experimenter-defined list. |
| **Fast vs. slow tempo** | |
| 80 bpm and below | When no in-text description of tempo was provided, tempo values were retrieved using the Spotify Web API (https://developer.spotify.com) and rounded to the nearest integer. |
| 81–100 bpm | |
| 101–120 bpm | |
| 121–140 bpm | |
| 141–160 bpm | |
| 161 bpm and above | |
| Unspecified | |
| **Stress induction procedure** | |
| With SET | Stress induction procedure included a social-evaluative threat (SET) component. |
| Without SET | Stress induction procedure did not include a social-evaluative threat component. |
| **Stress check** | |
| Successful | Stress induction procedure elicited an acute stress response. |
| Unsuccessful | Stress induction procedure did not elicit an acute stress response. |
| Unreported | Effect of stress induction procedure was not directly reported. |
| **Outcome type (general)** | |
| Neuroendocrine | Includes cortisol & salivary IgA. |
| Physiological | Includes heart rate, heart rate variability indices, systolic and diastolic blood pressure, respiration rate, skin conductance, and salivary alpha-amylase. |
| Psychological | Includes subjective stress, perceived relaxation, state anxiety, state depression, rumination, positive affect, and negative affect. |
| **Outcome type (specific)** | |
| Cortisol | |
| Salivary IgA | |
| Heart rate | |
| Heart rate variability indices: | |
| RMSSD | |
| LF | |
| HF | |
| LF/HF | |
| Entropy | |
| Systolic blood pressure | |

(*Continued*)

**Table 1.** (Continued)

| Moderator (bolded) and level | Criteria |
|---|---|
| Diastolic blood pressure | |
| Respiration rate | |
| Skin conductance | |
| Salivary alpha-amylase | |
| Subjective stress | |
| Anxiety | |
| State depression | |
| Relaxation | |
| Rumination | |
| Positive affect | |
| Negative affect | |
| **Duration of music** | Kept as a continuous moderator. |

**Test of overall effect and moderators.** To estimate the overall effect of music listening on stress recovery, we fit an intercept-only, random-effects meta-regression model with RVE using the 'robu' function of the 'robumeta' package [93]. The intercept estimated by this model can be interpreted as the precision-weighted overall effect size which has been corrected for dependencies. We used a similar approach to estimate cumulative effect sizes at each level of each moderator. For cases where a level of a moderator had too few observations for the RVE approach, we calculated cumulative effect sizes by fitting a random-effects meta-regression using the 'rma.mv' function of the 'metafor' package [88].

Prior to conducting moderator analyses, categorical moderators (e.g., 'Genre') were dummy coded, while the continuous moderator 'Duration' was left as is. For cases where the categorical moderator only had two levels, moderator variables were entered into separate meta-regression equations using the RVE approach. The significance test of the regression coefficient for the predictor variable in the meta-regression equation was interpreted as a test of whether the variable was a significant moderator. We used the same approach to test the effect of continuous moderators. For cases where the categorical moderator had more than two levels, moderator variables were entered into separate random-effects meta-regression models. This yielded $QM$ and $QE$ statistics: the $QM$ statistic indicated whether there was a significant difference among all levels of the tested moderator, while the $QE$ statistic indicated whether there were significant amounts of residual heterogeneity after accounting for the effect of the moderator [94].

**Publication bias.** The most common method to assess publication bias in meta-analytic data sets with dependent effect sizes is to aggregate individual effect sizes from the same study, and subsequently perform standard publication bias tests on the aggregated estimates. Therefore, we first aggregated individual effect sizes using the 'agg' function of the 'MAd' package [95]. The 'agg' function calculates aggregated effect size and variance estimates using formulas specified in Borenstein et al. [92]. These aggregated estimates were then used to assess publication bias by means of: (a) Egger's regression of funnel plot asymmetry [96]; (b) a trim-and-fill analysis [97]; and (c) PET-PEESE models [98].

## Results

Overall, the analyses comprised $s = 14$ studies, from which $k = 90$ effect sizes were calculated. The cumulative sample size of these studies was $N = 706$, while individual sample sizes ranged from 12–143 participants, with a mean of approximately 68 participants per study.

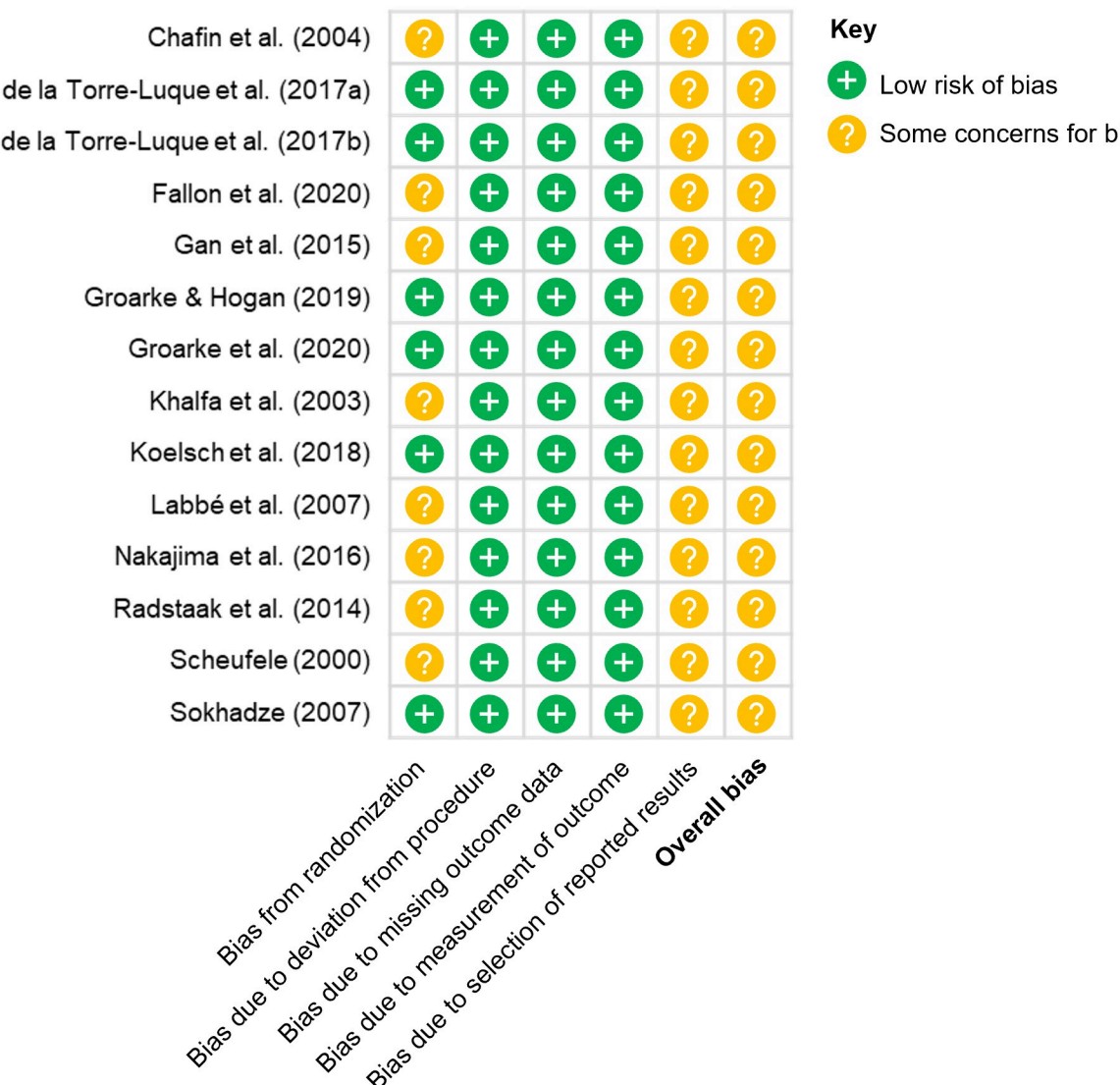

**Fig 2. Quality of included studies based on the RoB 2.** All studies were of overall moderate quality due to little-to-no information on pre-specification of analysis plans, making it difficult to fully rule out any bias that may occur due to selection of reported results.

## Overall effect

Based on a meta-regression with RVE, the estimated overall effect of music listening on stress recovery was $g = 0.15$, 95% CI [-0.21, 0.52], $t(13) = 0.92$, $p = 0.374$. This estimate suggests that, taking all variations in music and outcomes into consideration, the effect of music listening on stress recovery is equivalent to silence or an auditory control.

## Outlier detection

Using the 'influence' function of the 'metafor' package [88], one influential outlier in the negative direction was detected [18]. The overall effect of music listening on stress recovery with outlier removed was $g = 0.18$, 95% CI [-0.18, 0.54], $t(13) = 1.08$, $p = 0.300$. The full meta-analytic data set was retained in subsequent analyses.

**Table 2. Coded information and effect sizes of studies included in the meta-analysis (s = 14; k = 90).**

| Study | Year | Check | SIP | Selection | Genre | Lyrics | Tempo[†] | Duration[††] | Outcome Type | Outcome Measure | N | g |
|---|---|---|---|---|---|---|---|---|---|---|---|---|
| Chafin et al. | 2004 | Successful | With SET | Experimenter | Classical | Instrumental | 101–120 | 10 | Physiological | Heart rate | 30 | 0.47 |
| Chafin et al. | 2004 | Successful | With SET | Experimenter | Classical | Instrumental | 101–120 | 10 | Physiological | Systolic blood pressure | 30 | 1.53 |
| Chafin et al. | 2004 | Successful | With SET | Experimenter | Classical | Instrumental | 101–120 | 10 | Physiological | Diastolic blood pressure | 30 | 0.89 |
| Chafin et al. | 2004 | Successful | With SET | Experimenter | Jazz | Instrumental | > = 161 | 10 | Physiological | Heart rate | 30 | 0.17 |
| Chafin et al. | 2004 | Successful | With SET | Experimenter | Jazz | Instrumental | > = 161 | 10 | Physiological | Systolic blood pressure | 30 | 0.17 |
| Chafin et al. | 2004 | Successful | With SET | Experimenter | Jazz | Instrumental | > = 161 | 10 | Physiological | Diastolic blood pressure | 30 | 0.07 |
| Chafin et al. | 2004 | Successful | With SET | Experimenter | Pop | Lyrical | 101–120 | 10 | Physiological | Heart rate | 30 | 0.31 |
| Chafin et al. | 2004 | Successful | With SET | Experimenter | Pop | Lyrical | 101–120 | 10 | Physiological | Systolic blood pressure | 30 | 0.41 |
| Chafin et al. | 2004 | Successful | With SET | Experimenter | Pop | Lyrical | 101–120 | 10 | Physiological | Diastolic blood pressure | 30 | 0.23 |
| Chafin et al. | 2004 | Successful | With SET | Pseudo | Unspecified | Unspecified | Unspecified | 10 | Physiological | Heart rate | 30 | 0.35 |
| Chafin et al. | 2004 | Successful | With SET | Pseudo | Unspecified | Unspecified | Unspecified | 10 | Physiological | Systolic blood pressure | 30 | 0.36 |
| Chafin et al. | 2004 | Successful | With SET | Pseudo | Unspecified | Unspecified | Unspecified | 10 | Physiological | Diastolic blood pressure | 30 | 0.43 |
| De la Torre-Luque et al. | 2017a | Successful | With SET | Self | Unspecified | Unspecified | Unspecified | 16 | Physiological | Heart rate | 58 | -0.16 |
| De la Torre-Luque et al. | 2017a | Successful | With SET | Self | Unspecified | Unspecified | Unspecified | 16 | Physiological | RMSSD | 58 | -0.07 |
| De la Torre-Luque et al. | 2017a | Successful | With SET | Self | Unspecified | Unspecified | Unspecified | 16 | Physiological | LF | 58 | 0.44 |
| De la Torre-Luque et al. | 2017a | Successful | With SET | Self | Unspecified | Unspecified | Unspecified | 16 | Physiological | HF | 58 | -1.04 |
| De la Torre-Luque et al. | 2017a | Successful | With SET | Self | Unspecified | Unspecified | Unspecified | 16 | Physiological | LF/HF | 58 | -0.50 |
| De la Torre-Luque et al. | 2017a | Successful | With SET | Self | Unspecified | Unspecified | Unspecified | 16 | Physiological | Entropy | 58 | 0.57 |
| De la Torre-Luque et al. | 2017a | Successful | With SET | Self | Unspecified | Unspecified | Unspecified | 16 | Psychological | Anxiety | 58 | 0.57 |
| De la Torre-Luque et al. | 2017a | Successful | With SET | Self | Unspecified | Unspecified | Unspecified | 16 | Psychological | State depression | 58 | 0.61 |
| De la Torre-Luque et al. | 2017a | Successful | With SET | Self | Unspecified | Unspecified | Unspecified | 16 | Psychological | Positive affect | 58 | 0.45 |
| De la Torre-Luque et al. | 2017a | Successful | With SET | Self | Unspecified | Unspecified | Unspecified | 16 | Psychological | Negative affect | 58 | 0.66 |
| De la Torre-Luque et al. | 2017b | Successful | With SET | Experimenter | Unspecified | Instrumental | Unspecified | 15 | Physiological | Entropy | 21 | -0.62 |
| Fallon et al. | 2020 | Successful | With SET | Experimenter | Unspecified | Lyrical | < = 80 | 5 | Physiological | Skin conductance | 72 | 0.09 |
| Gan, Lim, & Haw | 2015 | Unreported | Without SET | Experimenter | Classical | Instrumental | 121–140 | 20 | Physiological | Heart rate | 70 | -0.17 |
| Gan, Lim, & Haw | 2015 | Unreported | Without SET | Experimenter | Classical | Instrumental | 121–140 | 20 | Physiological | Systolic blood pressure | 70 | 0.17 |

*(Continued)*

**Table 2.** (*Continued*)

| Study | Year | Check | SIP | Selection | Genre | Lyrics | Tempo[†] | Duration[††] | Outcome Type | Outcome Measure | N | g |
|---|---|---|---|---|---|---|---|---|---|---|---|---|
| Gan, Lim, & Haw | 2015 | Unreported | Without SET | Experimenter | Classical | Instrumental | 121–140 | 20 | Physiological | Diastolic blood pressure | 70 | 0.18 |
| Gan, Lim, & Haw | 2015 | Unreported | Without SET | Experimenter | Classical | Instrumental | > = 161 | 20 | Physiological | Heart rate | 70 | -0.26 |
| Gan, Lim, & Haw | 2015 | Unreported | Without SET | Experimenter | Classical | Instrumental | > = 161 | 20 | Physiological | Systolic blood pressure | 70 | -0.19 |
| Gan, Lim, & Haw | 2015 | Unreported | Without SET | Experimenter | Classical | Instrumental | > = 161 | 20 | Physiological | Diastolic blood pressure | 70 | 0.04 |
| Gan, Lim, & Haw | 2015 | Unreported | Without SET | Experimenter | Classical | Instrumental | 121–140 | 20 | Psychological | Anxiety | 70 | 0.60 |
| Gan, Lim, & Haw | 2015 | Unreported | Without SET | Experimenter | Classical | Instrumental | > = 161 | 20 | Psychological | Anxiety | 70 | 0.06 |
| Groarke & Hogan | 2019 | Successful | With SET | Self | Unspecified | Unspecified | Unspecified | 10 | Psychological | Subjective stress | 80 | 1.19 |
| Groarke & Hogan | 2019 | Successful | With SET | Self | Unspecified | Unspecified | Unspecified | 10 | Psychological | Anxiety | 80 | 1.68 |
| Groarke & Hogan | 2019 | Successful | With SET | Self | Unspecified | Unspecified | Unspecified | 10 | Psychological | Relaxation | 80 | 1.51 |
| Groarke & Hogan | 2019 | Successful | With SET | Self | Unspecified | Unspecified | Unspecified | 10 | Psychological | State depression | 80 | 0.12 |
| Groarke & Hogan | 2019 | Successful | With SET | Self | Unspecified | Unspecified | Unspecified | 10 | Psychological | Negative affect | 80 | 0.52 |
| Groarke et al. (study 1) | 2020 | Successful | With SET | Experimenter | Unspecified | Instrumental | < = 80 | 8 | Physiological | Systolic blood pressure | 46 | 0.28 |
| Groarke et al. (study 1) | 2020 | Successful | With SET | Experimenter | Unspecified | Instrumental | < = 80 | 8 | Physiological | Diastolic blood pressure | 46 | 0.01 |
| Groarke et al. (study 1) | 2020 | Successful | With SET | Experimenter | Unspecified | Instrumental | < = 80 | 8 | Psychological | Anxiety | 46 | 0.78 |
| Groarke et al. (study 1) | 2020 | Successful | With SET | Self | Unspecified | Unspecified | Unspecified | 10 | Physiological | Systolic blood pressure | 47 | 0.15 |
| Groarke et al. (study 1) | 2020 | Successful | With SET | Self | Unspecified | Unspecified | Unspecified | 10 | Physiological | Diastolic blood pressure | 47 | -0.03 |
| Groarke et al. (study 1) | 2020 | Successful | With SET | Self | Unspecified | Unspecified | Unspecified | 10 | Psychological | Anxiety | 47 | 0.84 |
| Groarke et al. (study 2) | 2020 | Successful | With SET | Experimenter | Unspecified | Instrumental | < = 80 | 8 | Physiological | Systolic blood pressure | 50 | -0.17 |
| Groarke et al. (study 2) | 2020 | Successful | With SET | Experimenter | Unspecified | Instrumental | < = 80 | 8 | Physiological | Diastolic blood pressure | 50 | -0.08 |
| Groarke et al. (study 2) | 2020 | Successful | With SET | Experimenter | Unspecified | Instrumental | < = 80 | 8 | Psychological | Anxiety | 50 | -0.39 |
| Groarke et al. (study 2) | 2020 | Successful | With SET | Self | Unspecified | Unspecified | Unspecified | 10 | Physiological | Systolic blood pressure | 50 | -0.22 |

(*Continued*)

**Table 2.** (Continued)

| Study | Year | Check | SIP | Selection | Genre | Lyrics | Tempo† | Duration†† | Outcome Type | Outcome Measure | N | g |
|---|---|---|---|---|---|---|---|---|---|---|---|---|
| Groarke et al. (study 2) | 2020 | Successful | With SET | Self | Unspecified | Unspecified | Unspecified | 10 | Physiological | Diastolic blood pressure | 50 | 0.01 |
| Groarke et al. (study 2) | 2020 | Successful | With SET | Self | Unspecified | Unspecified | Unspecified | 10 | Psychological | Anxiety | 50 | 0.09 |
| Khalfa et al. | 2003 | Successful | With SET | Self | Unspecified | Unspecified | Unspecified | 45 | Neuroendocrine | Cortisol | 17 | 1.20 |
| Koelsch et al. | 2016 | Successful | Without SET | Experimenter | Unspecified | Unspecified | 101–120 | 41 | Neuroendocrine | Cortisol | 143 | -1.10 |
| Labbé et al. | 2007 | Unreported | Without SET | Experimenter | Classical | Instrumental | Unspecified | 10 | Physiological | Heart rate | 28 | -0.01 |
| Labbé et al. | 2007 | Unreported | Without SET | Experimenter | Classical | Instrumental | Unspecified | 10 | Physiological | Respiration rate | 28 | 0.91 |
| Labbé et al. | 2007 | Unreported | Without SET | Experimenter | Classical | Instrumental | Unspecified | 10 | Physiological | Skin conductance | 28 | -0.09 |
| Labbé et al. | 2007 | Unreported | Without SET | Experimenter | Heavy Metal | Instrumental | Unspecified | 10 | Physiological | Heart rate | 28 | -0.12 |
| Labbé et al. | 2007 | Unreported | Without SET | Experimenter | Heavy Metal | Instrumental | Unspecified | 10 | Physiological | Respiration rate | 28 | 0.17 |
| Labbé et al. | 2007 | Unreported | Without SET | Experimenter | Heavy Metal | Instrumental | Unspecified | 10 | Physiological | Skin conductance | 28 | -0.28 |
| Gan, Lim, & Haw | 2015 | Unreported | Without SET | Experimenter | Classical | Instrumental | > = 161 | 20 | Physiological | Systolic blood pressure | 70 | -0.19 |
| Gan, Lim, & Haw | 2015 | Unreported | Without SET | Experimenter | Classical | Instrumental | > = 161 | 20 | Physiological | Diastolic blood pressure | 70 | 0.04 |
| Gan, Lim, & Haw | 2015 | Unreported | Without SET | Experimenter | Classical | Instrumental | 121–140 | 20 | Psychological | Anxiety | 70 | 0.60 |
| Gan, Lim, & Haw | 2015 | Unreported | Without SET | Experimenter | Classical | Instrumental | > = 161 | 20 | Psychological | Anxiety | 70 | 0.06 |
| Groarke & Hogan | 2019 | Successful | With SET | Self | Unspecified | Unspecified | Unspecified | 10 | Psychological | Subjective stress | 80 | 1.19 |
| Groarke & Hogan | 2019 | Successful | With SET | Self | Unspecified | Unspecified | Unspecified | 10 | Psychological | Anxiety | 80 | 1.68 |
| Groarke & Hogan | 2019 | Successful | With SET | Self | Unspecified | Unspecified | Unspecified | 10 | Psychological | Relaxation | 80 | 1.51 |
| Groarke & Hogan | 2019 | Successful | With SET | Self | Unspecified | Unspecified | Unspecified | 10 | Psychological | State depression | 80 | 0.12 |
| Groarke & Hogan | 2019 | Successful | With SET | Self | Unspecified | Unspecified | Unspecified | 10 | Psychological | Negative affect | 80 | 0.52 |
| Groarke et al. (study 1) | 2020 | Successful | With SET | Experimenter | Unspecified | Instrumental | < = 80 | 8 | Physiological | Systolic blood pressure | 46 | 0.28 |
| Groarke et al. (study 1) | 2020 | Successful | With SET | Experimenter | Unspecified | Instrumental | < = 80 | 8 | Physiological | Diastolic blood pressure | 46 | 0.01 |
| Groarke et al. (study 1) | 2020 | Successful | With SET | Experimenter | Unspecified | Instrumental | < = 80 | 8 | Psychological | Anxiety | 46 | 0.78 |
| Groarke et al. (study 1) | 2020 | Successful | With SET | Self | Unspecified | Unspecified | Unspecified | 10 | Physiological | Systolic blood pressure | 47 | 0.15 |

(*Continued*)

**Table 2.** (Continued)

| Study | Year | Check | SIP | Selection | Genre | Lyrics | Tempo† | Duration†† | Outcome Type | Outcome Measure | N | g |
|---|---|---|---|---|---|---|---|---|---|---|---|---|
| Groarke et al. (study 1) | 2020 | Successful | With SET | Self | Unspecified | Unspecified | Unspecified | 10 | Physiological | Diastolic blood pressure | 47 | -0.03 |
| Groarke et al. (study 1) | 2020 | Successful | With SET | Self | Unspecified | Unspecified | Unspecified | 10 | Psychological | Anxiety | 47 | 0.84 |
| Groarke et al. (study 2) | 2020 | Successful | With SET | Experimenter | Unspecified | Instrumental | < = 80 | 8 | Physiological | Systolic blood pressure | 50 | -0.17 |
| Groarke et al. (study 2) | 2020 | Successful | With SET | Experimenter | Unspecified | Instrumental | < = 80 | 8 | Physiological | Diastolic blood pressure | 50 | -0.08 |
| Groarke et al. (study 2) | 2020 | Successful | With SET | Experimenter | Unspecified | Instrumental | < = 80 | 8 | Psychological | Anxiety | 50 | -0.39 |
| Groarke et al. (study 2) | 2020 | Successful | With SET | Self | Unspecified | Unspecified | Unspecified | 10 | Physiological | Systolic blood pressure | 50 | -0.22 |
| Groarke et al. (study 2) | 2020 | Successful | With SET | Self | Unspecified | Unspecified | Unspecified | 10 | Physiological | Diastolic blood pressure | 50 | 0.01 |
| Groarke et al. (study 2) | 2020 | Successful | With SET | Self | Unspecified | Unspecified | Unspecified | 10 | Psychological | Anxiety | 50 | 0.09 |
| Khalfa et al. | 2003 | Successful | With SET | Self | Unspecified | Unspecified | Unspecified | 45 | Neuroendocrine | Cortisol | 17 | 1.20 |
| Koelsch et al. | 2016 | Successful | Without SET | Experimenter | Unspecified | Unspecified | 101–120 | 41 | Neuroendocrine | Cortisol | 143 | -1.10 |
| Labbé et al. | 2007 | Unreported | Without SET | Experimenter | Classical | Instrumental | Unspecified | 10 | Physiological | Heart rate | 28 | -0.01 |
| Labbé et al. | 2007 | Unreported | Without SET | Experimenter | Classical | Instrumental | Unspecified | 10 | Physiological | Respiration rate | 28 | 0.91 |
| Labbé et al. | 2007 | Unreported | Without SET | Experimenter | Classical | Instrumental | Unspecified | 10 | Physiological | Skin conductance | 28 | -0.09 |
| Labbé et al. | 2007 | Unreported | Without SET | Experimenter | Heavy Metal | Instrumental | Unspecified | 10 | Physiological | Heart rate | 28 | -0.12 |
| Labbé et al. | 2007 | Unreported | Without SET | Experimenter | Heavy Metal | Instrumental | Unspecified | 10 | Physiological | Respiration rate | 28 | 0.17 |
| Labbé et al. | 2007 | Unreported | Without SET | Experimenter | Heavy Metal | Instrumental | Unspecified | 10 | Physiological | Skin conductance | 28 | -0.28 |
| Labbé et al. | 2007 | Unreported | Without SET | Self | Unspecified | Unspecified | Unspecified | 10 | Physiological | Heart rate | 28 | 0.18 |
| Labbé et al. | 2007 | Unreported | Without SET | Self | Unspecified | Unspecified | Unspecified | 10 | Physiological | Respiration rate | 28 | 0.04 |
| Labbé et al. | 2007 | Unreported | Without SET | Self | Unspecified | Unspecified | Unspecified | 10 | Physiological | Skin conductance | 28 | -0.12 |
| Nakajima et al. | 2016 | Successful | Without SET | Experimenter | Classical | Instrumental | 81–100 | 4 | Physiological | Heart rate | 24 | 0.12 |
| Nakajima et al. | 2016 | Successful | Without SET | Experimenter | Classical | Instrumental | 81–100 | 4 | Physiological | LF | 24 | 0.88 |
| Nakajima et al. | 2016 | Successful | Without SET | Experimenter | Classical | Instrumental | 81–100 | 4 | Physiological | HF | 24 | 0.45 |
| Nakajima et al. | 2016 | Successful | Without SET | Experimenter | Classical | Instrumental | 81–100 | 4 | Physiological | LF/HF | 24 | 0.24 |

(*Continued*)

**Table 2.** (Continued)

| Study | Year | Check | SIP | Selection | Genre | Lyrics | Tempo[†] | Duration[††] | Outcome Type | Outcome Measure | N | g |
|---|---|---|---|---|---|---|---|---|---|---|---|---|
| Radstaak et al. | 2014 | Successful | With SET | Self | Unspecified | Unspecified | Unspecified | 5 | Physiological | Heart rate | 60 | 0.18 |
| Radstaak et al. | 2014 | Successful | With SET | Self | Unspecified | Unspecified | Unspecified | 5 | Physiological | Systolic blood pressure | 60 | -0.78 |
| Radstaak et al. | 2014 | Successful | With SET | Self | Unspecified | Unspecified | Unspecified | 5 | Physiological | Diastolic blood pressure | 60 | -0.41 |
| Radstaak et al. | 2014 | Successful | With SET | Self | Unspecified | Unspecified | Unspecified | 5 | Physiological | Heart rate | 62 | 0.00 |
| Radstaak et al. | 2014 | Successful | With SET | Self | Unspecified | Unspecified | Unspecified | 5 | Physiological | Systolic blood pressure | 62 | -0.51 |
| Radstaak et al. | 2014 | Successful | With SET | Self | Unspecified | Unspecified | Unspecified | 5 | Physiological | Diastolic blood pressure | 62 | -4.18 |
| Radstaak et al. | 2014 | Successful | With SET | Self | Unspecified | Unspecified | Unspecified | 5 | Psychological | Positive affect | 63 | 0.67 |
| Radstaak et al. | 2014 | Successful | With SET | Self | Unspecified | Unspecified | Unspecified | 5 | Psychological | Negative affect | 63 | 0.12 |
| Radstaak et al. | 2014 | Successful | With SET | Self | Unspecified | Unspecified | Unspecified | 5 | Psychological | Rumination | 63 | -0.45 |
| Radstaak et al. | 2014 | Successful | With SET | Self | Unspecified | Unspecified | Unspecified | 5 | Psychological | Positive affect | 65 | 0.96 |
| Radstaak et al. | 2014 | Successful | With SET | Self | Unspecified | Unspecified | Unspecified | 5 | Psychological | Negative affect | 65 | -0.03 |
| Radstaak et al. | 2014 | Successful | With SET | Self | Unspecified | Unspecified | Unspecified | 5 | Psychological | Rumination | 65 | -0.32 |
| Scheufele | 2000 | Successful | With SET | Experimenter | Classical | Instrumental | 81–100 | 15 | Physiological | Heart rate | 33 | 2.48 |
| Scheufele | 2000 | Successful | With SET | Experimenter | Classical | Instrumental | 81–100 | 15 | Psychological | Relaxation | 33 | -0.49 |
| Sokhadze | 2007 | Successful | Without SET | Experimenter | Classical | Instrumental | 121–140 | 2 | Physiological | Heart rate | 51 | -0.65 |
| Sokhadze | 2007 | Successful | Without SET | Experimenter | Classical | Instrumental | 121–140 | 2 | Physiological | HF | 51 | 0.26 |
| Sokhadze | 2007 | Successful | Without SET | Experimenter | Classical | Instrumental | 121–140 | 2 | Physiological | LF/HF | 51 | 0.46 |
| Sokhadze | 2007 | Successful | Without SET | Experimenter | Classical | Instrumental | 121–140 | 2 | Physiological | Skin conductance | 51 | -0.43 |
| Sokhadze | 2007 | Successful | Without SET | Experimenter | Classical | Instrumental | 81–100 | 2 | Physiological | Heart rate | 51 | 0.18 |
| Sokhadze | 2007 | Successful | Without SET | Experimenter | Classical | Instrumental | 81–100 | 2 | Physiological | HF | 51 | -0.18 |
| Sokhadze | 2007 | Successful | Without SET | Experimenter | Classical | Instrumental | 81–100 | 2 | Physiological | LF/HF | 51 | -0.27 |
| Sokhadze | 2007 | Successful | Without SET | Experimenter | Classical | Instrumental | 81–100 | 2 | Physiological | Skin conductance | 51 | 0.67 |
| Sokhadze | 2007 | Successful | Without SET | Experimenter | Classical | Instrumental | 121–140 | 2 | Psychological | Anxiety | 51 | -0.11 |
| Sokhadze | 2007 | Successful | Without SET | Experimenter | Classical | Instrumental | 81–100 | 2 | Psychological | Subjective stress | 51 | 0.06 |
| Sokhadze | 2007 | Successful | Without SET | Experimenter | Classical | Instrumental | 121–140 | 2 | Psychological | Anxiety | 51 | 0.08 |

(*Continued*)

**Table 2.** (Continued)

| Study | Year | Check | SIP | Selection | Genre | Lyrics | Tempo[†] | Duration[††] | Outcome Type | Outcome Measure | N | g |
|-------|------|-------|-----|-----------|-------|--------|-------|----------|--------------|-----------------|---|---|
| Sokhadze | 2007 | Successful | Without SET | Experimenter | Classical | Instrumental | 81–100 | 2 | Psychological | Subjective stress | 51 | 0.21 |

*Note*. A more detailed data file is available on the Open Science Framework.

[†] = In beats per minute (BPM);

[††] = in minutes.

Check = stress induction check/manipulation check; SIP = stress induction procedure; $N$ = total observations for two group comparison; $g$ = Hedges' $g$.

## Moderator analyses

There was significant heterogeneity of effect sizes ($T^2 = 0.71$, $I^2 = 89.29$) from each study, which suggests that meaningful differences may exist among studies that could be further explored through moderator analyses. Cumulative effect size estimates at each level of each moderator, along with their respective significance tests, are reported in Table 3.

**Classical music vs. other genres.** Our results suggest that the effect of music listening on stress recovery may differ across musical genres, $QM(4) = 27.19$, $p < .001$. Despite this, it is difficult to further elaborate on these differences as the individual estimated effects of pop ($g = 0.317$, 95% CI [0.09, 0.53], $p = .025$) and jazz music ($g = 0.137$, 95% CI [0.00, 0.27], $p = .049$) were derived from single studies, while the estimates for classical ($g = 0.431$, 95% CI [-0.03, 0.88], $p = .059$) and heavy metal music ($g = -0.076$, 95% CI [-0.64, 0.48], $p = .619$), along with music collapsed into the 'unspecified' category ($g = 0.067$, 95% CI [-0.42, 0.56], $p = .765$), were non-significant. Residual heterogeneity was statistically significant, $QE(67) = 1147.43$, $p < .001$.

**Instrumental vs. lyrical.** The effects of music listening on stress recovery did not differ between lyrical music ($g = 0.159$, 95% CI [-1.13, 1.45], $p = .362$), instrumental music ($g = 0.194$, 95% CI [-0.22, 0.65], $p = .273$), and music with 'unspecified' lyrical presence ($g = 0.151$, 95% CI [-0.46, 0.78], $p = .581$), $QM(2) = 3.44$, $p = .179$. Residual heterogeneity was statistically significant, $QE(69) = 1171.95$, $p < .001$.

**Self- vs. experimenter selected.** Our results suggest that there may be differences in magnitude between the effect of self-selected, pseudo self-selected, and experimenter selected music on stress recovery, $QM(2) = 19.13$, $p < .001$. However, these differences were difficult to expand on since the estimated effect of pseudo self-selected music (i.e., self-selected music from a list composed by experimenters) was derived from only one study ($g = 0.377$, 95% CI [0.27, 0.48], $p = .004$), while the estimated effects of self-selected ($g = 0.336$, 95% CI [-0.29, 0.96], $p = .226$) and experimenter selected music ($g = 0.030$, 95% CI [-0.33, 0.45], $p = .874$) were non-significant. Residual heterogeneity was statistically significant, $QE(69) = 1139.39$, $p < .001$.

**Fast vs. slow tempo.** Our results suggest that the effects of music listening on stress recovery may differ in magnitude based on musical tempo, $QM(5) = 43.66$, $p < .001$. However, little can be said about these differences since the estimated effects of music at 80 bpm or below ($g = 0.084$, 95% CI [-0.06, 0.23], $p = .086$), 81–100 bpm ($g = 0.497$, 95% CI [-0.62, 1.62], $p = .197$), 101–120 bpm ($g = -0.260$, 95% CI [-11.3, 10.8], $p = .815$), 121–140 bpm ($g = 0.067$, 95% CI [-1.58, 1.71], $p = .696$), 161 bpm and above ($g = -0.020$, 95% CI [-1.33, 1.29], $p = .870$), and 'unspecified' tempo ($g = 0.235$, 95% CI [-0.26, 0.73], $p = .301$) were non-significant. Residual heterogeneity was statistically significant, $QE(67) = 1128.90$, $p < .001$.

**Table 3. Moderator analyses.**

| Moderator (bolded) and level | s | k | g | $\beta_1$ | QM | 95% CI | p |
|---|---|---|---|---|---|---|---|
| **Classical vs. other genres** | 14 | 90 | - | - | 27.19 | - | < .001 |
| Classical | 6 | 32 | 0.431 | - | - | [-0.03, 0.88] | 0.059 |
| Heavy metal[†] | 1 | 3 | -0.076 | - | - | [-0.64, 0.48] | 0.619 |
| Jazz[†] | 1 | 3 | 0.137 | - | - | [0.00, 0.27] | 0.049 |
| Pop[†] | 1 | 3 | 0.317 | - | - | [0.09, 0.53] | 0.025 |
| Unspecified | 10 | 49 | 0.067 | - | - | [-0.42, 0.56] | 0.765 |
| **Instrumental vs. lyrical** | 14 | 90 | - | - | 3.44 | - | 0.179 |
| Instrumental | 8 | 45 | 0.194 | - | - | [-0.16, 0.55] | 0.240 |
| Lyrical[†] | 2 | 4 | 0.159 | - | - | [-1.13, 1.45] | 0.362 |
| Unspecified | 8 | 41 | 0.151 | - | - | [-0.46, 0.78] | 0.581 |
| **Self- vs. experimenter selected** | 14 | 90 | - | - | 19.13 | - | < .001 |
| Self | 6 | 37 | 0.336 | - | - | [-0.29, 0.96] | 0.226 |
| Experimenter | 10 | 50 | 0.030 | - | - | [-0.33, 0.45] | 0.874 |
| Pseudo[†] | 1 | 3 | 0.377 | - | - | [0.27, 0.48] | 0.004 |
| **Fast vs. slow tempo** | 14 | 90 | - | - | 43.66 | - | < .001 |
| 80 bpm and below | 2 | 7 | 0.084 | - | - | [-0.06, 0.23] | 0.086 |
| 81–100 bpm | 3 | 12 | 0.497 | - | - | [-0.62, 1.62] | 0.197 |
| 101–120 bpm | 2 | 7 | -0.260 | - | - | [-11.3, 10.8] | 0.815 |
| 121–140 bpm | 2 | 10 | 0.067 | - | - | [-1.58, 1.71] | 0.696 |
| 161 bpm and above | 2 | 7 | -0.020 | - | - | [-1.33, 1.29] | 0.870 |
| Unspecified | 8 | 47 | 0.235 | - | - | [-0.26, 0.73] | 0.301 |
| **Stress induction procedure** | 14 | 90 | - | -0.450 | - | [-1.22, 0.32] | 0.218 |
| With SET | 9 | 56 | 0.319 | - | - | [-0.15, 0.79] | 0.154 |
| Without SET | 5 | 34 | -0.141 | - | - | [-0.90, 0.62] | 0.636 |
| **Stress check** | 14 | 90 | - | -0.108 | - | [-1.47, 1.26] | 0.661 |
| Successful | 12 | 73 | 0.173 | - | - | [-0.26, 0.61] | 0.399 |
| Unsuccessful | 2 | 17 | 0.062 | - | - | [-0.08, 0.20] | 0.115 |
| **Outcome type (general)** | 14 | 90 | - | - | 164.22 | - | < .001 |
| Neuroendocrine | 2 | 2 | -0.004 | - | - | [-14.6, 14.6] | 0.998 |
| Physiological | 11 | 62 | 0.135 | - | - | [-0.39, 0.67] | 0.585 |
| Psychological | 7 | 26 | 0.298 | - | - | [-0.11, 0.71] | 0.127 |
| **Outcome type (specific)** | 14 | 90 | - | - | 374.12 | - | < .001 |
| Cortisol | 2 | 2 | -0.004 | - | - | [-14.6, 14.6] | 0.998 |
| Heart rate | 8 | 16 | 0.236 | - | - | [-0.40, 0.87] | 0.412 |
| Heart rate variability indices: | | | | | | | |
| RMSSD | 1 | 1 | -0.069 | - | - | [-0.58, 0.44] | 0.794 |
| LF | 2 | 2 | 0.562 | - | - | [-1.96, 3.08] | 0.216 |
| HF | 3 | 4 | -0.212 | - | - | [-2.12, 1.69] | 0.678 |
| LF/HF | 3 | 4 | -0.085 | - | - | [-1.11, 0.934] | 0.739 |
| Entropy | 2 | 2 | 0.031 | - | - | [-7.49, 7.55] | 0.967 |
| Systolic blood pressure | 4 | 12 | -0.040 | - | - | [-0.87, 0.74] | 0.880 |
| Diastolic blood pressure | 4 | 12 | -0.442 | - | - | [-2.39, 1.50] | 0.522 |
| Respiration rate | 1 | 3 | 0.362 | - | - | [-0.79, 1.51] | 0.309 |
| Skin conductance | 3 | 6 | 0.038 | - | - | [-0.29, 0.37] | 0.659 |
| Subjective stress | 2 | 3 | 0.665 | - | - | [-6.02, 7.35] | 0.426 |
| Anxiety | 5 | 10 | 0.579 | - | - | [-0.23, 1.39] | 0.118 |
| State depression | 2 | 2 | 0.345 | - | - | [-2.79, 3.48] | 0.395 |

*(Continued)*

**Table 3.** (Continued)

| Moderator (bolded) and level | s | k | g | β₁ | QM | 95% CI | p |
|---|---|---|---|---|---|---|---|
| Relaxation | 2 | 2 | 0.525 | - | - | [-12.2, 13.3] | 0.693 |
| Rumination | 1 | 2 | -0.383 | - | - | [-1.17, 0.41] | 0.102 |
| Positive affect | 2 | 3 | 0.636 | - | - | [-1.65, 2.92] | 0.176 |
| Negative affect | 3 | 4 | 0.404 | - | - | [-0.39, 1.19] | 0.155 |
| **Duration of music** | 14 | 90 | - | -0.005 | - | [-0.11, 0.10] | 0.870 |

Note.

† = moderator level contained too few observations to obtain an estimate using the RVE approach, so estimate was obtained by means of random-effects meta-regression.

$s$ = number of studies; $k$ = number of effect sizes; $g$ = Hedges' $g$. $β_1$ coefficients are from separate meta-regressions with RVE, where a categorical moderator with two levels was dummy coded and entered into the model as a predictor; $Q_M$ statistics are a Wald-type chi-square test which indicate whether there are significant differences among all levels of a moderator. The number of studies may not always add up, since most studies contributed multiple effect sizes.

**Stress induction procedure.** There were no significant differences in the effects of music listening on stress recovery between studies whose stress induction procedures included SET ($g$ = 0.319, 95% CI [-0.15, 0.79], $p$ = .154) and those without SET ($g$ = -0.141, 95% CI [-0.90, 0.62], $p$ = .636), $β_1$ = -0.450, $p$ = .218.

**Stress induction checks.** There were no significant differences in the effects of music listening on stress recovery for studies with successful ($g$ = 0.173, 95% CI [-0.26, 0.61], $p$ = .399) and unreported ($g$ = 0.062, 95% CI [-0.08, 0.20], $p$ = .115) stress induction checks, $β_1$ = -0.108, $p$ = .661.

**Type of outcome.** Our results suggest that the effects of music listening on stress recovery may differ between neuroendocrine, physiological, and psychological outcomes $QM(2)$ = 164.22, $p$ < .001. These differences were challenging to further expand on since the estimated effects of music listening for neuroendocrine ($g$ = -0.004, 95% CI [-14.6, 14.6], $p$ = .794), physiological ($g$ = 0.135, 95% CI [-0.39, 0.67], $p$ = .585), and psychological ($g$ = 0.298, 95% CI [-0.11, 0.71], $p$ = .127) stress recovery outcomes were not statistically significant. We noted a similar pattern when comparing the effects of music listening between *specific* stress recovery outcomes: the magnitude of the effect of music listening may vary across stress recovery outcomes, $QM(18)$ = 545.09, $p$ < .001, but estimated effects per outcome were non-significant (Table 3). Residual heterogeneity was statistically significant despite the inclusion of general outcome type ($QE(69)$ = 1018.57, $p$ < .001) and specific outcome measure ($QE(53)$ = 629.144, $p$ < .001) as moderators.

**Duration of music.** There was no evidence that the effect of music listening on stress recovery may differ depending on how long participants were exposed to music, $β_1$ = -0.005, $p$ = .870 ($range_{duration}$ = 2–45 minutes).

To further illustrate the methodological heterogeneity among experimental studies on the effect of music listening on stress recovery, we provide a more extensive, qualitative overview of the included studies in Appendix C. A summary of this overview is presented in Table 4.

## Publication bias

To visually assess the extent of publication bias, the aggregated effect size estimates in our meta-analytic data set were first used to create a plot of the estimates and their standard errors. In the absence of publication bias, this pattern should resemble a funnel, where effect size estimates with smaller standard errors cluster around the mean effect size, while effect size estimates with larger standard errors spread out in both directions. A common pattern which

**Table 4. Summary of studies included in the systematic review.**

| No. | Authors (Year) | N | Stress induction procedure | Music stimulus (Song [tempo]) | Measured outcomes | Reported findings |
|---|---|---|---|---|---|---|
| 1 | Chafin et al. (2004) | 75 | Arithmetic<br>*Description*: mental arithmetic with harassment.<br>Participants were asked to count back from a large, random number in odd steps (e.g., "Count backwards from 9000 in steps of 17") while being repeatedly interrupted (harassed) by the experimenter at timed intervals (e.g., "You are too slow, start over").<br>*Duration*: 5 minutes | Classical (Pachelbel–*Canon in D major*, [130 bpm]; Vivaldi–*The Four Seasons: Spring, Movement I*, [90 bpm])<br>Jazz (Miles Davis–*Flamenco Sketches*, [177 bpm])<br>Top 40 Pop (Sarah McLahlan–*Angel* [117 bpm]; Dave Matthews Band–*Crash Into Me*, [101 bpm])<br>Self-selected (Unspecified)<br>*Duration*: 10 minutes | *Physiological*<br>Systolic blood pressure<br>Diastolic blood pressure<br>Heart rate<br>*Psychological*<br>Anxiety (STAI, form A)<br>Rumination (1–7 scale)<br>Relaxation (1–7 scale) | Significant effect of music on systolic blood pressure, with classical music returning systolic blood pressure closer to baseline compared to control condition (+).<br>Similar pattern as systolic blood pressure, but not significant (-).<br>No significant differences between groups (-).<br>No significant differences between groups (-).<br>No significant differences between groups (-).<br>No significant differences between groups (-). |
| 2 | de la Torre-Luque et al. (2017a) | 21 | Modified Trier Social Stress Task (TSST)<br>*Description*: modified TSST with PASAT. Participants were asked to deliver a presentation in front of a camera, with the video feed and a timer displayed on a nearby laptop. The mental arithmetic component was substituted with the Paced Auditory Serial-Addition Task (PASAT). In the PASAT, participants are presented with a number every three seconds, and are asked to add the current presented number with the number presented before (Gronwall, 1977).<br>*Duration*: 15 minutes | Unspecified (Melomics relaxing music)<br>*Duration*: 16 minutes | *Physiological*<br>Heart rate variability (HR, RMSSD, LF, HF, LF/HF, SampEn)<br>*Psychological*<br>Anxiety (STAI) | Significant differences in HR, LF, HF, and SampEn at baseline and recovery phases. Music group demonstrated significantly higher SampEn during recovery phase compared to control group (+).<br>Significant difference in anxiety across study phases, but not between groups (-). |
| 3 | de la Torre-Luque et al. (2017b) | 58 | Modified TSST<br>*Description*: modified TSST with PASAT.<br>*Duration*: 15 minutes | Self-selected (Unspecified)<br>*Duration*: 15 minutes | *Physiological*<br>Heart rate variability (HR, RMSSD, LF, HF, LF/HF, SampEn)<br>*Psychological*<br>Anxiety (STAI)<br>Depression (ST-DEP)<br>Positive affect (PANAS)<br>Negative affect (PANAS) | Significant differences in HR, LF, LF/HF, and SampEn across study phases. Music group demonstrated significantly higher HF and SampEn during recovery phase compared to control group (+).<br>Anxiety scores for music group during recovery phase significantly lower compared to control group (+).<br>Depression scores for music group during recovery phase significantly lower compared to control group (+).<br>Positive affect for music group during recovery phase significantly higher compared to control group (+).<br>Negative affect for music group during recovery phase significantly lower compared to control group (+). |
| 4 | Fallon et al. (2020) | 105 | Modified TSST<br>*Description*: standard TSST with shorter mental arithmetic component.<br>*Duration*: 11 minutes | Unspecified (Eric Whitacre–*Sleep* [50 bpm])<br>*Duration*: 5 minutes | *Physiological*<br>Skin conductance<br>*Psychological*<br>Current mood (Irritated, Satisfied, Excited, Distracted, Tingling feeling, Calm) | Significant differences in skin conductance between study sessions (baseline, stressor, recovery). Significant differences in skin conductance between music listening group and silence control group during recovery session (+)<br>Significant differences in current mood between study sessions. Music listening intervention did not have differential effects on current mood compared to control group (-) |

*(Continued)*

**Table 4.** (Continued)

| No. | Authors (Year) | N | Stress induction procedure | Music stimulus (Song [tempo]) | Measured outcomes | Reported findings |
|---|---|---|---|---|---|---|
| 5 | Gan et al. (2016) | 105 | Arithmetic<br>*Description*: participants were asked to complete 12 questions from the University of Cambridge General Certificate of Education (GCE) Ordinary-level mathematics examinations.<br>*Duration*: 15 minutes | Stimulative (Beethoven–*Moonlight Sonata No.14 in C-sharp Minor Op. 72 No. 2*, [171 bpm])<br>Sedative (Camille Saints-Saens–*Allegro Moderato*, *Symphony No. 3 Op. 78*, *Movement III*, [121 bpm])<br>*Duration*: approx. 20 minutes | *Physiological*<br>Systolic blood pressure<br>Diastolic blood pressure<br>Heart rate<br>*Psychological*<br>Anxiety (STAI, form X-1)<br>Math anxiety (MARS) | No significant effect of music post-stressor. Significant difference in systolic blood pressure post-music compared to baseline for stimulative music, sedative music, and control groups (-).<br>No significant effect of music post-stressor. Significant difference in diastolic blood pressure post-music compared to baseline for sedative music and control groups (-). No significant differences between groups (-).<br>Significant decrease in post-stress anxiety scores for sedative music group compared to control group (+).<br>Significant decrease in post-stress mathematics anxiety scores for sedative music group compared to control group (+). |
| 6 | Groarke & Hogan (2019) | 40 | Modified TSST<br>*Description*: standard TSST with speech component omitted.<br>*Duration*: 10 minutes | Self-selected (Unspecified)<br>*Duration*: 10 minutes | *Psychological*<br>Subjective stress<br>Nervousness<br>Tension<br>Upset<br>Sadness<br>Depressed affect | Significant differences in subjective stress between music group and control group (+).<br>Significant differences in nervousness between music group and control group (+).<br>No significant differences in tension between study groups (-).<br>Significant differences in upset regulation between music group and control group (+).<br>Significant differences in sadness between music group and control group (+).<br>Significant differences in depressed affect between music group and control group (+). |
| 7 | Groarke et al. (2020) | 70 | Modified TSST<br>*Description*: standard TSST with speech component omitted.<br>*Duration*: 8 minutes | Unspecified (Marconi Union–*Weightless* [71 bpm]<br>Self-selected (Unspecified)<br>*Duration*: approx. 10 minutes | *Physiological*<br>Systolic blood pressure<br>Diastolic blood pressure<br>*Physiological*<br>Anxiety (STAI) | Significant changes in systolic blood pressure across study phases, but no significant differences in systolic blood pressure recovery between study conditions (-).<br>Significant changes in diastolic blood pressure across study phases, and no significant differences in diastolic blood pressure recovery between study conditions (-).<br>Significant differences in post-stressor anxiety for both music groups compared to control group (+). |
|  |  | 75 | Modified TSST<br>*Description*: standard TSST with speech component omitted.<br>*Duration*: 8 minutes | Unspecified (Marconi Union–*Weightless* [71 bpm]<br>Self-selected (Unspecified)<br>*Duration*: approx. 10 minutes | *Physiological*<br>Systolic blood pressure<br>Diastolic blood pressure<br>*Physiological*<br>Anxiety (STAI) | Significant changes in systolic blood pressure across study phases, but no significant differences in systolic blood pressure recovery between study conditions (-).<br>No significant changes in diastolic blood pressure across study phases, and no significant differences in diastolic blood pressure recovery between study conditions (-).<br>No significant differences in post-stressor anxiety between all study groups (-). |

*(Continued)*

**Table 4.** (*Continued*)

| No. | Authors (Year) | N | Stress induction procedure | Music stimulus (Song [tempo]) | Measured outcomes | Reported findings |
|---|---|---|---|---|---|---|
| 8 | Khalfa et al. (2003) | 17 | TSST *Description*: standard TSST. Participants were asked to deliver an impromptu, interview-style presentation in front of a panel of judges who do not provide feedback or encouragement. This presentation is followed by a surprise mental arithmetic task (Allen et al., 2017; Kirschbaum, Pirke, & Hellhammer, 1993). *Duration*: 15 minutes | Unspecified (Various songs by Enya, Vangelis, & Yanni) *Duration*: 45 minutes | *Physiological* Cortisol | Significant, rapid decrease in post-stressor cortisol in music group compared to control group (+). |
| 9 | Koelsch et al. (2016) | 143 | $CO_2$ Stress Task *Description*: participants were instructed to take a single, vital-capacity breath of air containing 35% carbon dioxide and 65% oxygen. The $CO_2$ Stress Task is known to provoke panic attacks in many individuals with panic disorder, and has recently been used in stress research as an acute physiological stressor (Vickers, Jafarpour, Mofidi, Rafat, & Woznica, 2012). *Duration*: n/a | Unspecified (Unspecified) *Duration*: approx. 41 minutes | *Physiological* Cortisol *Psychological* Mood (POMS, measures Depression/Anxiety, Fatigue, Vigor, Irritability) | Increase in post-stressor cortisol for music group significantly higher compared to control group (-). Significant increase in post-stressor positive mood scores in music group compared to control group (+). |
| 10 | Labbé et al. (2007) | 56 | Arithmetic *Description*: mental arithmetic operations were part of a broader "cognitive speed test" which also included number memory items, verbal analogy items, and spelling items. *Duration*: 10 minutes | Classical (Unspecified) Heavy metal (Unspecified) Self-selected (Unspecified) *Duration*: 20 minutes | *Physiological* Heart rate Respiration rate Skin conductance *Psychological* Relaxation (RSS) Anxiety (STAI, form Y) | No significant differences between groups (-). No significant differences between groups (-). Post-hoc, all groups experienced significant post-stressor decrease in skin conductance, which was larger for the classical and self-selected music groups (+). Relaxation scores for classical, self-selected, and silence groups significantly higher post-stressor compared to heavy metal group (-). Anxiety scores for classical and self-selected music groups significantly lower post-stressor compared to heavy metal and silence groups (+). |
| 11 | Nakajima et al. (2016) | 12 | Unpleasant stimuli *Description*: friction noise made by scratching a blackboard. *Duration*: 90 seconds | Classical (Mozart–*Horn Concerto No.2 in E flat major*, *Movement II*, [111 bpm]) *Duration*: approx. 4 minutes | *Physiological* Heart rate variability (HR, HFnu, LFnu, LF/HF) | HFnu significantly higher for music stimulus with amplified high frequency component, compared to music stimulus with amplified low frequency component (+). |
| 12 | Radstaak et al. (2014) | 123 | Arithmetic *Description*: mental arithmetic task with harassment. *Duration*: 5 minutes | Relaxing (Unspecified) Happy (Unspecified) *Duration*: 5 minutes | *Physiological* Systolic blood pressure Diastolic blood pressure Heart rate *Psychological* Positive affect (1–10 scale) Negative affect (1–10 scale) Rumination (1–10 scale) | Systolic blood pressure during recovery phase for relaxing music and happy music groups significantly higher compared to audio and silence control groups (-). No significant differences between groups (-). No significant differences between groups (-). Significant increase in positive affect during recovery phase for relaxing music and happy music groups compared to audio and silence control groups (+). No significant differences between groups (-). No significant differences between groups (-). |

(*Continued*)

**Table 4.** (Continued)

| No. | Authors (Year) | N | Stress induction procedure | Music stimulus (Song [tempo]) | Measured outcomes | Reported findings |
|-----|----------------|---|----------------------------|-------------------------------|-------------------|-------------------|
| 13 | Scheufele (2000) | 67 | Anticipation *Description*: faux presentation. *Duration*: 15 minutes | Classical (Mozart–*Sonata in D major for Two Pianos*, [100 bpm]]) *Duration*: 15 minutes | *Physiological* Heart rate *Psychological* Relaxation (VAS, *very tense-very relaxed*) Mood (POMS-SF, Tension subscale) | Significant differences in heart rate post-stressor for music group compared to attention control group (+). No significant differences between groups (-). No significant differences between groups (-). |
| 14 | Sokhadze (2007) | 29 | Unpleasant stimuli *Description*: nine pictures from the International Affective Picture System (IAPS; Lang, Bradley, & Cuthbert, 1997), which were presented to participants in series of three pictures. The nine IAPS pictures used in the study had been previously rated as strongly eliciting disgust (e.g., a mutilated body). *Duration*: 20 seconds per picture | Pleasant (*Spring Song*, [82 bpm]) Sad (Pachelbel–*Canon in D major*, [130 bpm]) *Duration*: 2 minutes | *Physiological* Electrodermal activity (SCL, SCR-M, NS.SCR) Heart rate variability (HR, LF, HF, LF/HF) *Psychological* Anxiety (1–7 scale) Depression (1–7 scale) Subjective stress (1–7 scale) | NS.SCR for pleasant music group significantly lower during music compared to during stressor (+). HF for pleasant music group significantly lower during music compared to during stressor. Post-stressor HF was significantly lower for pleasant music group compared to control group (-). No significant differences between groups (-). No significant differences between groups (-). No significant differences between groups (-). |

*Note.* (+) = Finding in support of the effect of music on physiological recovery from stress; (-) = Finding not in support of the effect of music on physiological recovery from stress.

suggests publication bias is asymmetry in the bottom of the plot. Fig 3 presents the funnel plot of the aggregated effect sizes.

Given the limited number of studies included in the meta-analysis ($n$ = 14), an accurate visual assessment of asymmetry was difficult. Thus, to supplement our visual inspection of the funnel plot, we conducted a trim-and-fill analysis, which trims the values of extreme estimates that may lead to asymmetry in the funnel plot and imputes values to balance out the distribution. No studies were imputed by the trim-and-fill analysis. Additionally, an Egger's regression for funnel plot asymmetry using the aggregated effect sizes failed to detect significant evidence of publication bias ($t$(12) = 1.26, $p$ = 0.231). Lastly, both PET ($\beta_1$ = 2.63, $p$ = 0.311) and PEESE ($\beta_1$ = 3.87, $p$ = 0.356) models were not statistically significant. Taken together, based on the aggregated effect sizes, the different methods of publication bias detection suggest that there is no evidence of publication bias. However, considering the small number of included studies and the significant heterogeneity of our meta-analytic data set, firm conclusions about the extent of publication bias within the current literature on the effects of music listening and stress recovery are difficult to make.

## Discussion

Music listening has the potential to fulfill the promise of effective stress recovery in healthy individuals. However, cumulative evidence from 17 experimental studies suggests that support for the beneficial effect of music listening on stress recovery is currently lacking: for healthy individuals, the effect of music listening on stress recovery may be equivalent to that of other auditory stimuli, or even merely sitting in silence. Furthermore, the effect of music listening on stress recovery is heterogeneous, and moderator analyses suggest the effect may differ in magnitude according to musical genre, whether music is self-selected, musical tempo, and

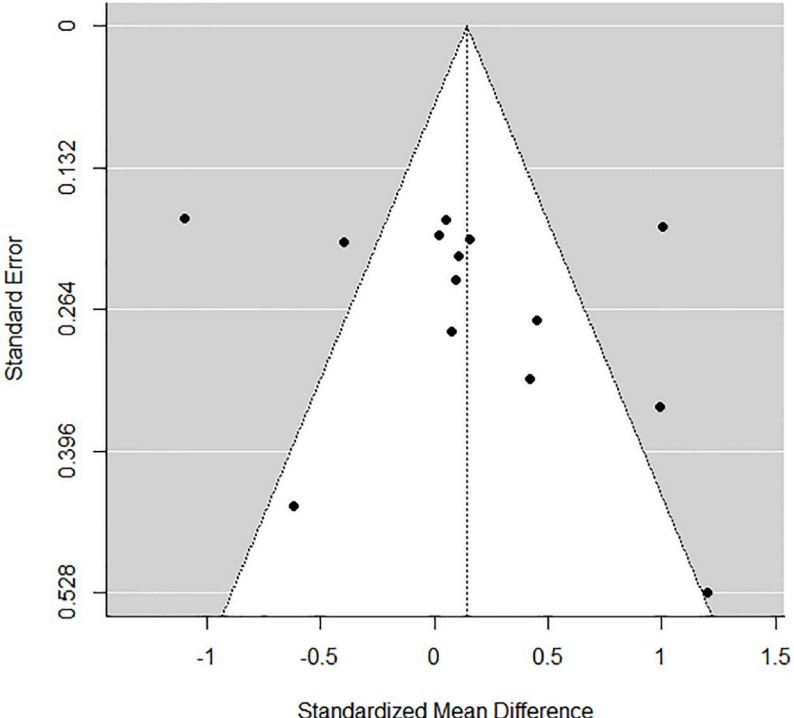

**Fig 3. Funnel plot of studies examining the effect of music listening on psychophysiological recovery from stress.** The small number of studies renders it difficult to visually inspect asymmetry, and thus precludes an accurate assessment of publication bias.

type of stress recovery outcome. Despite this, the limited number of available studies makes it difficult to draw further conclusions from these analyses.

## Overall effects of music listening on stress recovery

The results of our review contrast those of previous meta-analyses, which underscore the relevance of music-based interventions for stress-reduction [10, 11]. While previous reviews suggest that music-based interventions may be moderately beneficial for stress-related outcomes, particularly in medical and therapeutic settings, our results suggest that the magnitude of this effect outside of these settings, particularly for healthy individuals under acute, experimentally induced stress, may be more modest. We presume that one of the principal reasons for this difference was our decision to exclude studies conducted in medical and therapeutic settings. In previous reviews, randomized controlled trials of the effects of music-based interventions within medical and therapeutic settings constituted a large portion of included studies: 67 of 79 (85%) studies in de Witte et al. [10], and 15 of 22 (68%) studies in Pelletier [11], making it more likely that overall effect sizes were derived from studies conducted within these settings. Tentatively, the effects of music listening may be more prominent for the stress recovery of individuals in medical or therapeutic contexts, compared to that of individuals under acute stress in an experimental context. Whereas the time course of stress responses and stress recovery in experimental settings can be considered relatively brief [24, 26, 40, 83], the time course of stress responses and stress recovery within medical and therapeutic settings may be significantly more protracted [12, 13]. Thus, within medical and therapeutic settings, music may be exerting its influence on neuroendocrine, physiological, and psychological processes that have been subjected to longer periods of strain [27, 99].

Furthermore, the difference in overall estimated effect sizes may be attributed to differences in the breadth of music activities encompassed by our review and that of de Witte et al. [10]: whereas we included studies in which participants merely listened to music following a stressor, de Witte et al. [10] also included music therapy, along with other unspecified music activities. We speculate that the effect of music on stress recovery may differ depending on whether music is merely listened to, performed, or used within a music therapy setting. However, studies comparing the stress recovery effects of these various music activities are rare [15, 58]. Thus, future investigations into the differential effect of these music activities may therefore provide a more comprehensive picture of the effects of music on stress recovery.

## Potential moderating effects

Our review highlights the considerable methodological variety between studies investigating the effects of music listening on stress recovery. This is particularly concerning given the modest number of experimental studies on music listening and stress recovery in current literature. Although we investigated the impact of these methodological differences through moderator analyses, many of the estimated effects at each level of each moderator were either non-significant or originated from single studies. Taken together, meaningful interpretations for these moderating effects are difficult to make. Therefore, for each significant moderator, we instead provide several recommendations for future research, which we believe may help delineate the effects of these potential moderators.

**Musical genre.** Although comparisons between musical genres seem relatively straightforward, investigating the differential effects of musical genres may be particularly challenging: the conceptualization of musical genres, along with the songs they encompass, tends to be somewhat arbitrary [69, 75, 100, 101]. Indeed, studies display considerable variation in musical stimuli, even within the same genre (Table 4). A notable example of this is the study by Sandstrom and Russo [53], which utilized four 'classical' songs, each at different extremes of valence and arousal. It should also be considered that new music is continuously being released which may not completely fit with the definition of any existing genre [9].

As such, an alternative approach to the investigation of musical genre involves describing these genres according to their musical features, such as tempo, timbre, and loudness, and subsequently investigating the effects of these individual musical features on stress recovery [9, 101]. For example, classical music may be described as rhythmically complex, with mellow timbre and fluctuating loudness. Comparatively, though equally rhythmically complex, heavy metal possesses sharper timbre and more pronounced loudness. Investigating the differential effects of these musical features on stress recovery may provide relevant insight into the differential effects of listening to various musical genres on stress recovery.

**Self- versus experimenter selection.** In investigating the effects of self- versus experimenter selected music on stress recovery in healthy individuals, studies typically request participants to select music they consider 'relaxing' prior to an experiment [3, 17, 18]. Although this approach is viable, it precludes the potential role of perceived control in the relationship between music listening and stress recovery, since allowing participants to self-select their own music may already be helpful for stress recovery due to a restoration of perceived control [15]. Our results were not able to provide a significant contribution to this discussion, as hardly any experimental studies in our review have attempted to account for the potential effects of perceived control. As such, when contrasting the effects of self- and experimenter selected music on stress recovery, future studies may benefit from the inclusion of perceived control as an additional variable in their theoretical models.

It should also be noted that allowing participants to self-select their own music will result in a considerable variety of musical stimuli. Given that each of these musical stimuli may possess a different combination of musical features, the use of self-selected music may generate confounding effects that should preferably be accounted for. Arguably, self-selected music may produce consistent effects on stress-recovery regardless of underlying musical features, given that individuals tend to select music in service of personal self-regulatory goals [64, 75, 76]. However, given that variations in specific musical features, such as tempo, pitch, and loudness have been related to various physiological (e.g., heart rate) [73] and psychological stress recovery outcomes (e.g., positive and negative affect) [100–102], future studies may benefit from ensuring that musical features are consistent between self- and experimenter selected musical stimuli. This may be done, for instance, by comparing expert ratings of musical features [18]. Alternatively, there may be value in allowing participants to self-select music from a list provided by experimenters [21], as this would allow experimenters to standardize musical features a-priori, which may further help disentangle the effects of music listening from that of perceived control.

The comparison of musical features between self-selected and experimenter selected music may also offer a more nuanced perspective on the role of preference and familiarity. Specifically, preferences and familiarity towards certain songs could be described in terms of specific (combinations of) musical features. For example, an individual may prefer music with slow tempo, mellow timbre, and moderate loudness. This approach is often leveraged by music recommender systems, such as those implemented by music streaming platforms (e.g., Spotify, Deezer, Apple Music, etc.), with the goal of recommending songs that listeners are likely to engage with. Future studies could investigate the extent to which preference and familiarity might differ between self-selected and experimenter selected music with similar combinations of musical features, to further clarify the role of selection in the relationship between music listening and stress recovery.

**Musical tempo.** The systematic review portion of our results demonstrates that no studies have directly compared the effect of different musical tempi on stress recovery in healthy individuals. As such, the most straightforward approach to delineate the effects of musical tempo on stress recovery would be to adopt procedures in which participants listen to the same musical stimulus post-stressor, which is then varied in tempo across experimental conditions. Furthermore, even when the goal of a particular study on music listening and stress recovery is not to clarify the effects of musical tempo, we suggest that tempo values for each musical stimulus should be noted down and reported, as this would facilitate the comparison of the differential effects of musical tempo on stress recovery in future meta-synthesis of the literature.

Alternatively, the notion that music with slow tempo is more beneficial for stress recovery compared to music with fast tempo is supported by the assumption that physiological parameters will entrain to musical rhythms [63, 68]. As such, a more accurate approach to investigate the effects of musical tempo on stress recovery would be to leverage the dynamic, temporal nature of both music and physiological parameters through use of non-linear analyses of continuous data [52, 103]. For example, cross-recurrence quantification analysis (CRQA) [104, 105] may enable future studies to quantify the magnitude and duration of rhythmic entrainment for each participant. These indexes of magnitude and duration could then be compared between different musical tempi. Studies have utilized CRQA to investigate cardiac entrainment between participants of collective rituals [106] and the entrainment of an audience's heart rate to a live musical performance [107]. This analytical approach may therefore yield a more nuanced understanding of the effect of musical tempo on the recovery of autonomic parameters.

**Stress recovery outcomes.**   During short-term stress responses, catecholamine- and cortisol-mediated stress responses follow temporally specific patterns: catecholamines rapidly exert their influence on ANS activity, and these changes tend to normalize within 30–60 minutes [26]. Meanwhile, decreases in cortisol that may be attributed to stress recovery will only become noticeable after recovery-related changes in autonomic activity have begun to occur [24]. As such, to further clarify the effect of music listening on various stress recovery outcomes, we recommend future studies to be more sensitive towards the innate, intricate, and temporally specific changes of each stress recovery outcome.

Furthermore, multiple studies included in our review have opted to analyze continuous data by means of multivariate analyses of variance, after averaging participants' observed stress recovery outcomes at multiple time points (e.g., pre-stress, post-stress, post-recovery). Although this approach is practical, doing so may over-simplify the complex changes that may occur during the stress response and subsequent stress recovery, such as the temporal dynamics of different physiological responses [52] and emotion regulation strategies [108]. As such, we again suggest future studies to utilize non-linear analyses of data when appropriate, particularly when investigating the effects of music listening on the recovery of autonomic activity post-stressor. The idea of using non-linear analyses, such as time-series analysis, to investigate the stress recovery process is not new [5]. However, few studies on music listening and stress recovery have utilized this analytical approach.

## Additional recommendations

Two studies with unreported stress induction procedures were still included in the review [17, 84], as reported means for certain recovery outcomes still suggested an increase from baseline that participants could recover from. For example, with the information reported in Gan et al. [84], assuming a correlation of 0.5 between baseline and post-stressor measures of state anxiety, we estimated that their stress induction procedure elicited a significant increase in state anxiety in their sedative music ($t(34) = 5.87$, $p < .001$, $m_{diff} = 8.17$, $SD_{diff} = 8.24$), stimulative music ($t(34) = 8.21$, $p < .001$, $m_{diff} = 12.42$, $SD_{diff} = 8.95$), and control ($t(34) = 13.15$, $p < .001$, $m_{diff} = 15.83$, $SD_{diff} = 7.12$) conditions. As the overall estimated effect of music listening on the recovery process of healthy individuals following laboratory stressors may be relatively modest, it becomes particularly important to ensure that a sufficient stress response is elicited, to provide a larger window of opportunity in which the effect of music listening may be exerted on participants' recovery processes. We thus encourage future studies to adopt validated, (variations of) well-known stress tasks, such as the TSST [109], SECPT [110], or CO2 stress task [111], which have been demonstrated to consistently elicit marked physiological and psychological stress-related responses in laboratory settings. Furthermore, we remind future studies to candidly report the results of their stress induction procedures to facilitate subsequent meta-syntheses of the effects of music listening on stress recovery.

As the current review focused on the effects of music listening *after* a stressor, studies where music was played before or during a stressor were omitted from our analyses. However, several studies suggest that the timing at which music is played (i.e., before, during, or after a stressor) may influence its effects on stress recovery. For example, in Burns et al. [48], participants who listened to classical music while anticipating a stressful task exhibited lower post-music heart rate compared to participants who anticipated the stressor in silence. Similarly, concentrations of salivary cortisol were lower for participants who watched a stressful visual stimulus while listening to music compared to those who watched the same stimulus without music [112]. Together, these findings hint that, when listened during a stressor, music may attenuate cortisol responses [9, 113], thus reducing the subsequent need for recovery. On the

other hand, Thoma et al. [9] reported that participants who listened to music prior to a stressor exhibited higher post-stressor cortisol compared to participants who listened to an audio control. Interestingly, despite the stronger stress response, Thoma et al. [9] noted a trend for quicker ANS recovery among participants who listened to music, particularly with regards to salivary alpha-amylase activity. This pattern of findings is consistent with the notion forwarded by Koelsch et al. [61], in that music listening may promote a more adaptive stress response, thus facilitating subsequent stress recovery processes. To date, research on timing differences in the context of music listening and stress recovery is scarce. Thus, future studies could further examine the influence of such timing differences to better understand their role in the relationship between music listening and stress recovery.

Given the pervasiveness of stress, Ecological Momentary Assessment (EMA) studies may provide a more intimate outlook on the dynamics of daily music listening behaviour, particularly for the purpose of stress recovery. For example, through an ambulatory assessment study, Linnemann et al. [38] revealed that music produced the most notable reductions in physiological and psychological stress outcomes when it was listened to for the purpose of 'relaxation', compared to other reasons such as 'distraction', 'activation', and 'reducing boredom'. Indeed, given their high ecological validity, EMA studies may provide further insight into important contextual variables in the relationship between music listening and stress recovery. For example, in an EMA study, listening to music in the presence of others was related to decreased subjective stress, attenuated cortisol secretion, and higher activity of salivary alpha-amylase [55]. Furthermore, physiological responses to music may co-vary between members of a dyad when music is listened to by couples [114]. Thus, given the benefits of EMA studies, we invite future studies to continue exploring the dynamics and contextual factors of music listening behaviour for stress recovery in daily life.

Lastly, we encourage studies to support open science research practices, and to clearly report statistical information that may be relevant for meta-syntheses (e.g., means and standard deviations per time point, per experimental condition, etc.). Additionally, based on our assessment of study quality using the RoB 2, pre-registration of analysis plans can be helpful to ensure that the conducted study is of overall high quality. Next, we encourage studies to note down which specific musical stimuli were used, particularly those self-selected by participants [69, 99], as this enables future exploratory analyses of structural commonalities between different musical stimuli. Musical features from individual songs may be extracted by means of audio information extraction packages, such as MIRtoolbox [115]. Alternatively, individual song titles may be used to query related meta-data from online databases of various music streaming platforms. This meta-data can subsequently be used to obtain additional insight into the effects of music listening on stress recovery.

## Limitations of the current review

To our knowledge, our review is the first to comprehensively investigate the effect of music listening on stress recovery within healthy individuals. Given the explicit focus of our review, our meta-analytic data set excluded the more prominent effects of music listening in both medical and therapeutic settings [12, 13], allowing us to obtain results that are tentatively more representative of daily stress recovery processes. Despite this, the present review is not without its limitations:

First, although the specific focus of our review has allowed us to obtain a portrait of the effects of music listening on stress recovery in well-controlled experimental settings, the results of our review may be difficult to generalize to situations in which individuals experience prolonged stress responses. Stress induction procedures in experimental studies are designed to elicit acute

stress responses that are meant to subside upon conclusion of an experiment [83]. Although we believe these procedures provide a suitable approximation of typical stressors in daily life, certain stressors in daily life may also persist for a longer time. The manner and magnitude in which music listening influences prolonged stress responses may potentially differ from the way music influences acute, laboratory-induced stress responses [18, 45]. However, studies investigating the effect of music listening on stress recovery in the long-term are particularly rare.

Next, despite our best efforts to obtain relevant meta-analytic information from all studies selected for our review, our meta-analytic data set was ultimately constructed from a subset of fourteen studies. Although the subset allowed us to extract sufficient information to estimate an overall effect of music listening on stress recovery, several estimated effects at moderator level were derived from merely one or two studies (see Table 3). This precluded us from drawing further, meaningful conclusions about the results of our moderator analyses.

Finally, despite our clear focus on the effects of music listening on stress recovery within healthy individuals, there was substantial heterogeneity in our meta-analytic data set that could not be fully explained by the inclusion of moderators. Although the systematic review portion of our results highlighted potential additional sources of between-study heterogeneity, these additional sources could not be evaluated in our meta-analytic data set. We note, for example, that all studies utilized different musical stimuli to investigate the effect of music listening on stress recovery (see Table 4). The differential effects of these musical stimuli were difficult to account for in our meta-analysis, given the limited number of included studies. Overall, the significant heterogeneity in our meta-analytic data set suggests that our moderator analyses should be interpreted with caution.

## Conclusion

Studies commonly suggest that listening to music may have a positive influence on stress recovery. Based on cumulative evidence from 90 effect sizes in 14 studies, it may be premature to firmly conclude whether music listening is beneficial for the stress recovery of healthy individuals. The present review underscores the necessity for further and finer research into the effects of music, bearing the potential role of various moderators, such as musical genre, self-selection, musical tempo, and different stress recovery outcomes, to fully comprehend the nuanced effects of music listening on short-term stress recovery.

## Appendix A

### Search strategy

Using the advanced search feature within RUQuest, Web of Science, and PsycINFO, the following syntax was used so that the search returned results if keywords were found within the title, abstract, or keywords of relevant publications:

ti: (*music** OR "*music listening*") AND ((*stress** OR *strain* OR *recover** OR *relax** OR *fatigue* OR "*heart rate*" OR "*heart rate variability*" OR "*blood pressure*" OR *cardiovascular* OR *physiological* OR *cortisol* OR "*perseverative cognition*" OR *ruminat** OR *detachment* OR *distract** OR *worry** OR *emotion** OR *affect** OR *mood* OR *burnout* OR *depress**) NOT (*patient* OR *disease* OR *surgery* OR *operating* OR *theat*?? OR *disorder* OR *clinical* OR *stroke* OR *animal* OR *dent** OR *material* OR *recogni** OR *recommend**))

### OR

ab: (*music** OR "*music listening*") AND ((*stress** OR *strain* OR *recover** OR *relax** OR *fatigue* OR "*heart rate*" OR "*heart rate variability*" OR "*blood pressure*" OR *cardiovascular* OR

*physiological* OR *cortisol* OR "*perseverative cognition*" OR *ruminat** OR *detachment* OR *distract** OR *worry** OR *emotion** OR *affect** OR *mood* OR *burnout* OR *depress**) NOT (*patient* OR *disease* OR *surgery* OR *operating* OR *theat*?? OR *disorder* OR *clinical* OR *stroke* OR *animal* OR *dent** OR *material* OR *recogni** OR *recommend**))

## OR

kw: (*music** OR "*music listening*") AND ((*stress** OR *strain* OR *recover** OR *relax** OR *fatigue* OR "*heart rate*" OR "*heart rate variability*" OR "*blood pressure*" OR *cardiovascular* OR *physiological* OR *cortisol* OR "*perseverative cognition*" OR *ruminat** OR *detachment* OR *distract** OR *worry** OR *emotion** OR *affect** OR *mood* OR *burnout* OR *depress**) NOT (*patient* OR *disease* OR *surgery* OR *operating* OR *theat*?? OR *disorder* OR *clinical* OR *stroke* OR *animal* OR *dent** OR *material* OR *recogni** OR *recommend**))

## Appendix B

### Exploratory moderator analysis with study quality

Based on the RoB 2, all studies in the meta-analysis were of moderate quality, since the lack of pre-specified analysis plans from included studies made it difficult to completely rule out bias from the selection of reported results. Exploratorily, we conducted a less stringent assessment of study quality assuming all studies contained no bias due to selection of results. Based on this assessment, 7 (50%) of the included studies were high quality, while the remaining were moderate quality.

Following our procedure for moderator analyses, we conducted an additional random-effects meta-regression with RVE to test whether the estimated effect of music listening on stress recovery was stable across studies of different quality. The meta-regression suggests that study quality is a significant moderator of the effect of music listening on stress recovery, $QM$ (1) = 41.95, $p < .001$. The estimated effect of music listening on stress recovery in high quality studies was $g = 0.178$, 95% CI [0.00, 0.35], $p = .046$, while the estimated effect of music in moderate quality studies was $g = 0.102$, 95% CI [-0.14, 0.35], $p = .041$.

## Appendix C

### Stress induction procedures

In our meta-analysis, we generally distinguished between stress induction procedures with- or without a socio-evaluative threat component. However, specific stress induction procedures varied considerably between studies, as described below:

**Arithmetic tasks.** Four studies utilized arithmetic tasks to induce stress in participants. These tasks included single- and double-digit mental arithmetic operations [17], mental arithmetic operations "with harassment" [18, 21], and standardized mathematic tests [84].

**Trier Social Stress Task (with modifications).** One study [54] followed the standard administration protocol of the Trier Social Stress Task (TSST) [109, 116]. Two studies modified the TSST [109] by having participants prepare and deliver their presentations in front of a camera instead of a panel of judges [3, 37], while the subsequent mental arithmetic task was replaced by the Paced Auditory Serial Addition Test (PASAT) [117], administered through a laptop. One study administered the TSST with a shorter mental arithmetic component [118], while two studies omitted the TSST's speech delivery component [119, 120].

**Anticipation.** One study made use of anticipation to induce stress [50], where participants were asked to prepare an impromptu presentation that would be videotaped at the end

of a preparation period. Participants were eventually not required to deliver the prepared presentation.

**Unpleasant stimuli.**   Two studies exposed participants to unpleasant stimuli as a means of inducing stress. These unpleasant stimuli were either auditory [82] or visual [19] in nature.

**$CO_2$ stress task.**   One study utilized the $CO_2$ Stress Task [61]. In this task, as a an acute physiological stressor, participants were instructed to take a single, vital-capacity breath of air containing 35% carbon dioxide and 65% oxygen [111].

The duration of each stress induction procedure varied according to procedure category. The longest stress induction procedures (15 minutes) typically involved (variations of) the TSST (e.g., [37]. Conversely, the shortest procedure (90 seconds) was the exposure to unpleasant noise in Nakajima et al. [82], as their experimental design involved repeated presentation of the stressor to participants. Finally, it is also worth noting that among studies which reported successful stress induction procedures (see Table 2), the magnitude of resulting stress responses was often not reported.

## Selection of musical stimuli

All studies held a general assumption that 'relaxing' music would best promote stress recovery. However, studies utilized different strategies in selecting 'relaxing' music, resulting in considerable variation in musical stimuli between studies. These strategies are listed below:

**Sampling from available music.**   Four studies utilized a relatively straightforward strategy in selecting music—musical stimuli were sampled from songs commonly found on 'relaxing', either from their inclusion in anti-stress cassettes [21, 54], coverage in popular media [120], or the researcher's opinion [118].

**Referencing prior studies.**   Three studies selected music that, in prior studies, seemed to have positive effects on heart rate, respiration rate, perceived arousal, and perceived relaxation. One study made reference to pilot studies [82], while the remaining two cited previous published work by the same authors [19, 50].

**Theoretical conceptualization.**   Two studies attempted to theoretically conceptualize which music would be 'relaxing', and selected their musical stimuli accordingly. De la Torre-Luque et al. [3] utilized Melomics, a computational system for the automatic composition of music, to create songs that would be considered 'relaxing'. These songs were slow-paced, instrumental pieces, which contained no sudden or abrupt changes in melody. Gan et al. [84] distinguished between stimulative and sedative ('relaxing') music based on musical tempo— the speed or pace of a given song, and dynamic range—the difference between the quietest and loudest parts of a song [121]. In their study, stimulative music was characterized by fast tempo and broad dynamic range, while sedative music was characterized by slow tempo and narrow (soft) dynamic range.

**Self-selection.**   Six studies allowed participants to select and listen to their own 'relaxing' music. In four studies, participants were instructed to bring a list of 'preferred' relaxing music, which they would have the opportunity to listen to during the study [18, 37, 54, 66, 119]. In one study, participants selected 'relaxing' music from a list created by the experimenters (*pseudo* self-selection) [21]. The specific musical stimuli chosen by participants in studies allowing self-selection were often not reported.

## Effects of music listening on stress recovery

Studies utilized a variety of outcomes to investigate the effects of music listening on stress recovery. To expand upon the results of our meta-analysis, we detail the findings reported for each of these outcomes below. Given that three studies included in the systematic review could

not be included in the meta-analysis due to incomplete reported data, the number of studies per outcome reported in this section may differ from the number of studies per outcome in the meta-analysis (Table 3).

**Heart rate.** Scheufele [50] reported that participants who listened to classical music demonstrated lower post-stressor heart rate (HR) compared to participants in a comparable control group. By contrast, six studies reported no significant differences in post-stressor HR between participants who listened to music and those who did not [3, 18, 19, 21, 66, 84]. In summary, only one study out of seven provides evidence in support of a positive effect of music listening on post-stressor HR recovery.

**Heart rate variability.** Four studies utilized various heart rate variability (HRV) indices as a means to assess stress recovery. Two studies reported higher post-stressor HF band power in participants who listened to music compared to those who sat in silence [3, 37]. In Nakajima et al. [82], this difference was more pronounced for participants who listened to music with boosted high frequencies. Contrarily, in Sokhadze [19], participants who listened to peaceful music demonstrated lower post-stressor HF band power compared to those who sat in silence. Two studies reported that post-stressor sample entropy was higher for participants who listened to music compared to silence [3, 37]. This difference was taken as indicator which suggested that the physiological parameters of participants in the music condition were more ready to change compared to those in the silence condition [3]. No studies reported significant differences in RMSSD, LF band power, and LF/HF ratio between participants who listened to music and those who did not [3, 19, 37, 82]. Overall, three of four studies provide support for a positive effect of music listening on post-stressor HRV recovery, but these effects seem to vary across HRV indices.

**Blood pressure.** Four studies assessed the impact of music listening on stress recovery through changes in systolic blood pressure (SBP) and diastolic blood pressure (DBP). Chafin et al. [21] reported that the post-stressor SBP approached baseline values more quickly for participants who listened to experimenter-selected classical music compared to participants who sat in silence. On the other hand, three studies reported no significant differences in post-stressor SBP between participants who listened to music and those who did not [18, 84, 120]. Instead, compared to participants sitting in silence, post-stressor SBP recovery in participants who listened to either happy or relaxing music was delayed [18]. With regards to DBP, none of the above studies reported significant differences in post-stressor DBP between their respective experimental conditions. In summary, one study out of four provides support for a beneficial effect of music listening on post-stressor SBP recovery, while no studies provide support for a beneficial effect of music listening on DBP recovery.

**Respiration rate.** One study reported no significant differences in post-stressor respiration rate (RR) between participants listening to different musical genres and silence [17]. As such, there is currently no evidence to suggest that music listening is beneficial for post-stressor RR recovery.

**Skin conductance.** In Sokhadze [19], participants' SC was lower while listening to pleasant music compared to during the stressor. In Fallon et al. [118], participants who listened to self-selected music experienced lower SC compared to those in the control condition during the recovery session of the study. In a post-hoc analysis, Labbé et al. [17] reported that post-stressor SC recovery was greater for the classical and self-selected music groups, compared to the heavy metal or no music groups. Collectively, three studies provide evidence for a positive effect of music listening on post-stressor SC recovery.

**Cortisol.** Two studies utilized cortisol to examine the effect of music listening on stress recovery. Khalfa et al. [54] reported that post-stressor cortisol decreased more rapidly for participants who listened to experimenter-selected classical music, compared to participants who

sat in silence. Contrarily, Koelsch et al. [61] reported that music listening delayed cortisol recovery, as cortisol concentrations were higher for participants who listened to music post-stressor compared to silence. As such, only one study out of two provides support for a beneficial effect of music listening on post-stressor cortisol recovery.

**Subjective stress.**   In Groarke & Hogan [119], participants who listened to self-selected music reported lower subjective stress post-stressor compared to those who listened to a radio documentary. By comparison, in Radstaak et al. [18], there were no differences in post-stressor subjective stress between participants listening to happy music, relaxing music, an audiobook, and silence. Thus, only one study out of two provides support for a beneficial effect of music listening on post-stressor subjective stress.

**Perceived relaxation.**   In Labbé et al. [17], post-stressor perceived relaxation was higher for participants who listened to classical music compared to heavy metal, but not compared to silence. There were no significant differences in post-stressor perceived relaxation between participants listening to the various musical genres in Chafin et al. [21], and between participants listening to classical music or silence [50]. Thus, no studies provide conclusive evidence that music listening is beneficial for post-stressor perceived relaxation. However, the effects of music listening on perceived relaxation may differ depending on genre.

**State anxiety.**   Three studies reported that music listening reduced post-stressor state anxiety compared to silence [17, 37, 119]. Furthermore, Gan, Lim, and Haw [84] reported that post-stressor changes in mathematics-related anxiety were significantly higher for participants who listened to sedative music compared to those who did not. Despite this, three studies reported no significant differences in post-stressor state anxiety between their respective experimental groups [3, 19, 21]. Thus, four of seven studies provide support for a beneficial effect of music listening on post-stressor state anxiety.

**State depression.**   Two studies looked at the presence and/or severity of depressive symptoms in order to assess whether or not music facilitated psychological recovery [19, 37]. However, only de la Torre-Luque et al. [37] reported significant positive differences in post-stressor depressive symptoms between participants who listened to music and those who did not.

**Rumination.**   Two studies measured rumination as an indicator of psychological stress recovery, and both reported no significant differences in post-stressor rumination between participants in their respective experimental conditions [18, 21]. As such, there is currently no evidence to suggest that music listening is beneficial for post-stressor rumination.

**Positive and negative affect.**   De la Torre-Luque et al. [37] noted that participants who listened to music reported higher positive affect scores and lower negative affect scores post-stressor compared to the control group. Similarly, Radstaak et al. [18] reported that participants who listened to happy or relaxing music reported higher post-stressor positive affect compared to participants who did not listen to music, but found no significant differences in post-stressor negative affect. Two studies utilized the Profile of Moods Scale (POMS) to assess post-stressor changes in affect. Koelsch et al. [61] noted that participants who listened to music demonstrated higher post-stressor POMS scores (suggesting higher positive affect) compared to those who sat in silence. On the other hand, Scheufele [50] reported no significant differences in post-stressor POMS scores between experimental groups. Two studies [118, 119] measured affect by asking participants to report whether they felt various emotions (e.g., calmness, nervousness) throughout the study. Fallon et al. [118] reported that music listening did not have differential effects on affect compared to silence, while Groarke and Hogan [119] noted that participants who listened to music demonstrated less negative affect (as indicated by lower scores on the various emotions that participants were asked to rate) compared to those who did not. Collectively, the effect of music listening on post-stressor positive and negative affect seemed to be mixed. Three studies provide support for the beneficial role of music

listening on post-stressor positive affect, and two studies provide support for the beneficial effect of music listening for negative affect.

## Supporting information

**S1 Checklist.**
(DOC)

## Author Contributions

**Conceptualization:** Krisna Adiasto, Debby G. J. Beckers, Madelon L. M. van Hooff, Karin Roelofs, Sabine A. E. Geurts.

**Data curation:** Krisna Adiasto.

**Formal analysis:** Krisna Adiasto.

**Investigation:** Krisna Adiasto.

**Methodology:** Krisna Adiasto, Debby G. J. Beckers, Madelon L. M. van Hooff, Karin Roelofs, Sabine A. E. Geurts.

**Project administration:** Krisna Adiasto.

**Supervision:** Debby G. J. Beckers, Madelon L. M. van Hooff, Karin Roelofs, Sabine A. E. Geurts.

**Validation:** Krisna Adiasto.

**Visualization:** Krisna Adiasto.

**Writing – original draft:** Krisna Adiasto.

**Writing – review & editing:** Debby G. J. Beckers, Madelon L. M. van Hooff, Karin Roelofs, Sabine A. E. Geurts.

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
