## [Decision Letter · Decision Letter 0]

22 Oct 2021

PONE-D-21-22806Music listening and stress recovery in healthy individuals: A systematic review with meta-analysis of experimental studiesPLOS ONE

Dear Dr. Adiasto,

Thank you for submitting your manuscript to PLOS ONE. After careful consideration, we feel that it has merit but does not fully meet PLOS ONE’s publication criteria as it currently stands. Therefore, we invite you to submit a revised version of the manuscript that addresses the points raised during the review process.

We look forward to receiving your revised manuscript.

Kind regards,

Urs M Nater

Academic Editor

PLOS ONE

2. We note that this manuscript is a systematic review or meta-analysis; our author guidelines therefore require that you use PRISMA guidance to help improve reporting quality of this type of study. Please upload copies of the completed PRISMA checklist as Supporting Information with a file name “PRISMA checklist”.

Reviewers' comments:

Reviewer's Responses to Questions

**Comments to the Author**

1. Is the manuscript technically sound, and do the data support the conclusions?

Reviewer #1: Yes

Reviewer #2: Partly

2. Has the statistical analysis been performed appropriately and rigorously? 

Reviewer #1: Yes

Reviewer #2: Yes

3. Have the authors made all data underlying the findings in their manuscript fully available?

Reviewer #1: Yes

Reviewer #2: Yes

4. Is the manuscript presented in an intelligible fashion and written in standard English?

Reviewer #1: Yes

Reviewer #2: Yes

5. Review Comments to the Author

Reviewer #1: This systematic review and meta-analysis examined the effects of music listening after experimentally induced stress on stress recovery in healthy participants. The authors found no evidence for a cumulative effect of music listening on stress recovery. They did find that the effectiveness of music was moderated by several factors, including musical genre, type of music selection, musical tempo, and type of stress recovery outcome, although definite conclusions on the nature of these effects could not be drawn.

The study addresses a very timely question within the growing body of research on music and stress, and is well-executed. As such, it provides a valuable and much-needed contribution to the research field. The selection of study parameters and moderators of interest is convincing, the results are presented in a clear and comprehensible way, and the authors provide a thoughtful interpretation of their findings, which they appropriately put into perspective by acknowledging the limitations of their study. Furthermore, they provide several helpful and well-considered recommendations for future research. The manuscript is well-written.

I only have several minor comments.

1. In discussing the potential moderating effect “Self- vs. experimenter selected” (page 9), the authors mention two presumed explanations for this effect, namely increasing perceived control and serving self-regularity goals. For a somewhat more comprehensive picture, it may be worth adding the potential roles of liking and familiarity as further mechanisms behind the suggested higher effectiveness of self-compared to experimenter-selected music in promoting stress recovery.

2. In the abstract (and throughout the theoretical sections of the paper), it is stated that participants of the studies included in the meta-analysis/review “were either exposed to music or silence.” I find this misleading, since in the Method section, it is stated on page 10 that to be included, “studies should compare music listening to silence or a comparable auditory stimulus (e.g., white noise, audiobooks)”.

Apart from the fact that it is not evident in what sense and to what extent silence can be considered comparable to auditory control stimuli, the use of the label “silence” to capture all non-music control conditions, is confusing. Please adapt the instances where you currently refer to silence by using more accurate wording (e.g. “silence or an auditory control condition”).

3. Page 11: “When authors did not or could not provide the required information (e.g., due to data no longer being accessible), the outcome was dropped from the meta-analysis. Based on these criteria, the final sample for the systematic review consisted of 17 studies. Following attempts to obtain missing information, the final sample for the meta-analysis consisted of 14 studies.”

This is phrased in a confusing way – it is not clear what the conceptual difference between these selection steps is. Please rephrase this in a way that makes it less confusing.

4. On page 11, the authors point out that “Stress induction procedures in included studies were not always successful. Given that successful stress induction procedures are crucial to ensure that participants experience some physiological or psychological change they may recover from, in our moderator analysis we examined whether the effect of music listening on stress recovery differed based on the outcome of a study’s stress induction check (manipulation check)”.

I fully agree with the authors that, for music to exert an effect on stress, a physiological and/or psychological stress response needs to be present, from which participants may then recover. I find it therefore difficult to understand why studies which failed to induce stress (i.e. did not report a successful stress induction) were included in the meta-analysis in the first place. The fact that the successfulness of the stress induction, surprisingly, did not affect the extent of stress recovery does not really resolve my concern.

Could the authors briefly comment on this issue, and motivate their decision to still include these studies in their meta-analysis (either under “stress induction checks” on page 11, or in the discussion section)?

5. Page 11: “In our moderator analysis, we examined whether the effects of music listening on stress recovery were reliable across general (neuroendocrine, physiological, psychological) and specific outcome types.”

I am not sure whether the moderator analysis allows any claims about the reliability of the effects across outcome types. In theory, an effect could be highly reliable across many outcome types, while at the same time still being clearly stronger for some outcome types than for others (hence being moderated by them), right? Wouldn’t it be more correct to state that it was assessed to what extent the size of the effect on stress recovery depended on outcome type (or some equivalent formulation)? I am no expert on this issue, but I invite the authors to reconsider their wording.

6. There is a type on page 15: Wisagreements  Disagreements

7. As the authors rightly point out on page 6, stress recovery involves a process in which “changes that have occurred in response to a stressor revert to pre-stress baselines”. To quantify stress recovery, it therefore seems crucial to take individual pre-stress baseline levels into account.

To the reader, it does not readily become clear whether the effects derived from the studies included in the meta-analysis indeed reflect the extent to which stress levels “return to baseline”. From Table 4, the included studies seem to be a mix of 1) studies reporting differences in change scores with respect to pre-stress baseline levels and 2) studies reporting raw group differences in post-music stress levels. This may require some sort of disclaimer.

Could you please reflect on these analytical differences and their (possible) implications for the interpretation of your meta-analysis, in relation to the term “recovery”?

8. On page 37-38 you write: “Khalfa et al. [55] reported that post-stressor cortisol decreased more rapidly for participants who listened to experimenter-selected classical music, compared with participants who sat in silence”

In Table 2 you write, when referring to this study: “Increase in post-stressor cortisol for music group significantly lower compared to control group (+)”.

These descriptions differ – could you please adapt the main text to match the (correct) description in the table?

9. On page 40-41, you write: “While previous reviews suggest that music-based interventions may be moderately beneficial for stress-related outcomes, particularly in medical and therapeutic settings, our results suggest that the magnitude of this effect for healthy individuals may be more modest.”

While I largely agree with the contents of this paragraph (and with the further comments on this issue on page 46), I think the term “healthy individuals” (to label the category for which music is less effective for stress recovery) does not capture the essence of the differences between the different types of studies, and hence using this term may be a bit misleading.

As is stated further down the paragraph, the stress in studies conducted in medical and therapeutic settings likely has a more protracted time course, which does not directly have to do with the participants being (not) healthy. Furthermore, stress may differ in intensity between laboratory and medical real-life/settings, and the effectiveness of music may depend on the research setting as well.

It would be great if you could somewhat adapt the wording of this paragraph, to avoid the impression that the (non-) effectiveness of music depends on the participants being healthy. Rather, it seems more likely that several (interrelated) factors associated with the different research settings (e.g. type, intensity and duration of stress) are driving these differences. You might e.g. use the term “healthy individuals under brief, experimentally induced stress” instead.

Reviewer #2: Review:

The authors address an important research question as they aim at systemizing the empirical evidence on beneficial effects of music listening on stress recovery. Overall, the manuscript is well written and the authors demonstrate methodological rigour and diligence on many instances. However, I have some major concerns that question the adequacy of the hypothesis and statistical approach as well as the search strategy.

Major Concerns:

1) The authors identified 14 studies that are quite heterogenous in nature. I ask myself whether the approach of a meta-analysis is adequate for this rather small number of studies given this vast heterogenity. Furthermore, not all studies were successfull in stress induction. Wouldn't it be reasonable to exclude these studies from the analysis?

2) Concerning the search strategy, I wondered that PubMed was not included. Searching this data base might be useful as the number of studies identified is quite small.

3) Inclusion and exclusion criteria are not specifically justified. For example, in Figure 1 exclusion criteria are presented, e.g. 'no music presented after stressor'. This might be the reason why the study of Thoma et al. (2013) (https://pubmed.ncbi.nlm.nih.gov/23940541/) is not party of the review, although I consider it highly relevant in the context of music listening and stress reduction. Overall, I would describe and justify more in detail criteria for inclusion and exclusion of studies.

4) Introduction: first paragraph: I really like the introduction to the topic, as the aspect of stress in daily life is emphasized. I wondered why the authors did not include ambulatory assessment/ecological momentary assessment studies in their review, as these studies have high ecological validity. I recommend to expand the review and to assess independently the evidence concerning controlled studies with high internal validity on the one hand and daily life studies with high ecological validity on the other hand.

5) It may be my personal opinion, but I was irritated by the many instances the authors use the term 'beliefs', e.g. abstract 'given the popular and widespread belief'. My recommendation is to re-word this and acknowledge the empirical evidence underlying this statement.

Minor Concerns:

Abstract: 'beneficial for stress' should be specified (beneficial for stress reduction)

Abstract: Please report how many participants in total were included in the 14 studies

Abstract: please specify that (randomized)-controlled studies were included

Introduction: 'It is a popular and widespread opinion that music may be beneficial for

stress recovery [10]': I am not convinced that the citation is that adequate in this context. Levitin demonstrates in this book many instances for beneficial effects of music. I am not satisfied with labeling his statement as 'popular and widespread opinion'. Furthermore, I do not consider this book an optimal citation for this peer-reviewed journal article as there is a vast body of empirical evidence available.

Introduction l.60: My first thought was to question the necessity of this review given the fact that an extensive review was just published. Although, the authors justify their review in the ensuing paragraph, I would recommend to state immediately more clearly that the scope of the other review was different.

Introduction l.78: Why not include studies demonstrating effects on cortisol?

l.133: I would not recommend to write 'smaller amounts of salivary alpha-amylase'. Rather, less activity of alpha-amylase.

l.153: Study 54 refers to an ambulatory assessment study – therefore, there was no control to silence or noise in a comparable manner to experimental studies. Please re-word.

l.164: I do not agree with the statement that the candidate mechanism underlying beneficial effects of music has already been identified. I would rather prefer to see here a more comprehensive statement acknowledging that the exact underlying mechanisms remain to be elucidated and that different notions exist, e.g., literature by Koelsch…

l.274: IgA is named as outcome measure but has not been introduced. As it is an immune marker, the introduction should contain some information on interactions among stress and immune system.

l.310: some typos need revision

l.425: Can you please indicate the range of music duration? As there is literature available on the effects of different music durations on beneficial effects, I assume that the range was very limited among these studies. Therefore, I would not state that duration is not important. Rather, that the range in experimental studies is not vast enough to allow for meaningful comparisons.

6. PLOS authors have the option to publish the peer review history of their article (what does this mean?). If published, this will include your full peer review and any attached files.

Reviewer #1: No

Reviewer #2: No

---

## [Author Response · Author response to Decision Letter 0]

17 Dec 2021

Dear Prof. Dr. Nater,

Thank you for your e-mail in response to our submission to PLOS ONE [PONE-D-21-22806], in which you explained your decision concerning our manuscript, enclosing the reviewers’ comments. We have carefully addressed each reviewer’s comments and adjusted our manuscript accordingly. Our responses to each reviewer’s comments have been provided in a point-by-point manner below.

We hope the revised version of our manuscript will fully meet PLOS ONE’s rigorous publication criteria. We look forward to hearing from you.

Warm regards,

Krisna Adiasto, MSc.

On behalf of all authors.

---------------

Response to Reviewer #1

Dear Reviewer,

Thank you for taking the time to review our manuscript. We appreciate your kind words about our work. More importantly, we sincerely appreciate the constructive comments and suggestions you have provided for various portions of our review and have thus updated our manuscript to address them. Below, we provide a summary of the changes we have made. In the response letter we have attached to our revision, you will see your comments in bold, our responses in regular text, and excerpts from our manuscript in italics (unless otherwise indicated).

1. In discussing the potential moderating effect “Self- vs. experimenter selected” (page 9), the authors mention two presumed explanations for this effect, namely increasing perceived control and serving self-regularity goals. For a somewhat more comprehensive picture, it may be worth adding the potential roles of liking and familiarity as further mechanisms behind the suggested higher effectiveness of self-compared to experimenter-selected music in promoting stress recovery.

We agree that preference and familiarity potentially play a role in making self-selected music more beneficial compared to experimenter selected music for the purpose of stress recovery. As we strive to make our review as comprehensive as possible, we have added the following sentences on the roles of preference and familiarity in our discussion of the potential moderating effects of self- vs. experimenter selected music (page 9):

“Furthermore, previous studies have found that listening to self-selected music may help elicit stronger and more positive emotional responses regardless of a song’s valence (positive or negative) and arousal (high or low), possibly due to increased preference and familiarity towards the self-selected music (Jiang et al., 2016; Pereira et al., 2011; Sharman & Dingle, 2015). In theory, self-selected music should thus be more beneficial compared to experimenter-selected music for the purpose of stress recovery.”

And again in our discussion of the moderator analyses (page 45):

“The comparison of musical features between self-selected and experimenter selected music may also offer a more nuanced perspective on the role of preference and familiarity. Specifically, preferences and familiarity towards certain songs could be described in terms of specific (combinations of) musical features. For example, an individual may prefer music with slow tempo, mellow timbre, and moderate loudness. This approach is often leveraged by music recommender systems, such as those implemented by music streaming platforms (e.g., Spotify, Deezer, Apple Music, etc.), with the goal of recommending songs that listeners are likely to engage with. Future studies could investigate the extent to which preference and familiarity might differ between self-selected and experimenter selected music with similar combinations of musical features, to further clarify the role of selection in the relationship between music listening and stress recovery.”

Jiang, J., Rickson, D., & Jiang, C. (2016). The mechanism of music for reducing psychological stress: Music preference as a mediator. The Arts in Psychotherapy, 48, 62-68. https://doi.org/10.1016/j.aip.2016.02.002

Pereira, C. S., Teixeira, J., Figueiredo, P., Xavier, J., Castro, S. L., & Brattico, E. (2011). Music and emotions in the brain: Familiarity matters. PLOS ONE, 6(11), e27241. https://doi.org/10.1371/journal.pone.0027241

Sharman, L., & Dingle, G. A. (2015). Extreme metal music and anger processing. Frontiers in Human Neuroscience, 9, 272. https://doi.org/10.3389/fnhum.2015.00272

2. In the abstract (and throughout the theoretical sections of the paper), it is stated that participants of the studies included in the meta-analysis/review “were either exposed to music or silence.” I find this misleading, since in the Method section, it is stated on page 10 that to be included, “studies should compare music listening to silence or a comparable auditory stimulus (e.g., white noise, audiobooks)”.

Apart from the fact that it is not evident in what sense and to what extent silence can be considered comparable to auditory control stimuli, the use of the label “silence” to capture all non-music control conditions, is confusing. Please adapt the instances where you currently refer to silence by using more accurate wording (e.g., “silence or an auditory control condition”).

Thank you for your suggestion. It was not our intention to use the label ‘silence’ as an overall term for all control conditions across studies. When the change was justified (i.e., when the cited studies indeed had an auditory control condition), we have adjusted the instances in which use of the label ‘silence’ only was not accurate. For example, in the Abstract:

“As such, to clarify the current literature, we conducted a systematic review with meta-analysis of randomized, controlled experimental studies investigating the effects of music listening on stress recovery in healthy individuals. In fourteen experimental studies, participants (N = 706) were first exposed to an acute laboratory stressor, following which they were either exposed to music or a control condition.”

On page 4:

“Indeed, studies on music listening and stress recovery in healthy individuals are equivocal: although music listening is considered beneficial for physiological stress recovery, several studies have reported no differences in heart rate, heart rate variability, respiration rate, or blood pressure recovery between participants who listened to music and those who either sat in silence or listened to an auditory control [18-21].”

On page 7:

“For example, music listening has been associated with lower heart rate [49-51], systolic blood pressure [22, 50, 52], skin conductance [18, 20, 53, 54], and cortisol [55] compared to silence or an auditory control condition.”

“Furthermore, studies have demonstrated that listening to music may influence mood [60, 61]. Indeed, music listening has been associated with lower negative affect [38], higher positive affect [19, 62], and fewer self-reported depressive symptoms [38] compared to silence or an auditory control condition.”

On page 16:

“In the present study, a Hedges’ g of zero indicates the effect of music listening on stress recovery is equivalent to silence or an auditory control. Conversely, a Hedges’ g greater than zero indicates the degree to which music listening is more effective than control, while a g less than zero indicates the degree to which music listening is less effective than control.”

On page 22:

“This estimate suggests that, taking all variations in music and outcomes into consideration, the effect of music listening and silence have equivalent effects on stress recovery is equivalent to silence or an auditory control.”

3. Page 11: “When authors did not or could not provide the required information (e.g., due to data no longer being accessible), the outcome was dropped from the meta-analysis. Based on these criteria, the final sample for the systematic review consisted of 17 studies. Following attempts to obtain missing information, the final sample for the meta-analysis consisted of 14 studies.”

This is phrased in a confusing way – it is not clear what the conceptual difference between these selection steps is. Please rephrase this in a way that makes it less confusing.

Thank you for pointing this out. To make the distinction between the two steps clearer, we have rephrased the sentences in question to:

“Based on these criteria, the final sample for the systematic review portion of our manuscript consisted of 17 studies. Finally, for studies to be included in the meta-analysis portion of our review, means and standard deviations of stress recovery outcomes following stressor cessation must be available. Corresponding authors were contacted when this information was not available. When authors did not or could not provide the required information (e.g., due to data no longer being accessible), the outcome was dropped from the meta-analysis. Thus, following attempts to obtain missing information, the final sample for the meta-analysis portion of our review consisted of 14 studies.”

4. On page 11, the authors point out that “Stress induction procedures in included studies were not always successful. Given that successful stress induction procedures are crucial to ensure that participants experience some physiological or psychological change they may recover from, in our moderator analysis we examined whether the effect of music listening on stress recovery differed based on the outcome of a study’s stress induction check (manipulation check)”.

I fully agree with the authors that, for music to exert an effect on stress, a physiological and/or psychological stress response needs to be present, from which participants may then recover. I find it therefore difficult to understand why studies which failed to induce stress (i.e., did not report a successful stress induction) were included in the meta-analysis in the first place. The fact that the successfulness of the stress induction, surprisingly, did not affect the extent of stress recovery does not really resolve my concern.

Could the authors briefly comment on this issue and motivate their decision to still include these studies in their meta-analysis (either under “stress induction checks” on page 11, or in the discussion section)?

Thank you for mentioning this. We agree that it is particularly important to further address the inclusion of studies with unsuccessful stress induction checks, as you and another reviewer have put forward similar concerns on the matter.

In the previous version of our manuscript, three studies were coded to have ‘unsuccessful’ stress induction checks. One of these studies (Scheufele, 2000) was erroneously coded, as the author did report a marked increase in heart rate following their stress induction procedure compared to baseline. Meanwhile, in the remaining two studies (Gan et al., 2016 & Labbé et al., 2007), the authors have hinted that their respective stress induction procedures were successful. However, we coded these as ‘unsuccessful’ since the statistical analyses comparing post-stressor and baseline values of their stress recovery outcomes were missing. We ultimately decided to still include these two studies in the meta-analysis given that the reported mean scores for certain stress recovery outcomes still suggested there was an increase from baseline from which participants could recover from. For example, in Gan et al. (2016), mean state anxiety for their three conditions during the stress task were msedative = 46.97, mstimulative = 50.51, and mcontrol = 52.00, compared to baseline means scores of msedative = 38.80, mstimulative = 38.09, and mcontrol = 36.17. 

Based on this, we have thus updated the results of our analysis for the “stress induction checks” moderator (page 27), to reflect the change in coding for Scheufele (2000):

“Stress induction checks. There were no significant differences in the effects of music listening on stress recovery for studies with successful (g = 0.17301, 95% CI [-0.2635, 0.6155], p = .399625) and unsuccessful (g = 0.062361, 95% CI [-0.0894, 0.201.66], p = .115355) stress induction checks, β1 = -0.108257, p = .525661.”

Next, we have added a paragraph to acknowledge the inclusion of studies with unsuccessful stress tasks in our Discussion (page 49):

“Two studies with less successful stress induction procedures were still included in the review, given that reported raw scores for certain recovery outcomes still suggested an increase from baseline that participants could recover from. For example, in Gan et al. (2016), mean state anxiety for their three conditions during the stress task were msedative = 46.97, mstimulative = 50.51, and mcontrol = 52.00, compared to baseline means scores of msedative = 38.80, mstimulative = 38.09, and mcontrol = 36.17. Given that the overall estimated effect of music listening on the recovery process of healthy individuals following laboratory stressors may be relatively modest, it becomes particularly important to ensure that a sufficient stress response is elicited, to provide a larger window of opportunity in which the effect of music listening may be exerted on participants’ recovery processes. We thus encourage future studies to adopt validated (variations of) well-known stress tasks, such as the TSST, SECPT, or CO2 stress task, which have been demonstrated to consistently elicit marked physiological and psychological stress-related responses in laboratory settings.”

Gan, S. K. E., Lim, K. M. J., & Haw, Y. X. (2016). The relaxation effects of stimulative and sedative music on mathematics anxiety: A perception to physiology model. Psychology of Music, 44(4), 730-741. https://doi.org/10.1177/0305735615590430

Labbé, E., Schmidt, N., Babin, J., & Pharr, M. (2007). Coping with stress: the effectiveness of different types of music. Applied Psychophysiology and Biofeedback, 32, 163-168. https://doi.org/10.1007/s10484-007-9043-9

Scheufele, P. M. (2000). Effects of progressive relaxation and classical music on measurements of attention, relaxation, and stress responses. Journal of Behavioral Medicine, 23, 207-228. https://doi.org/10.21236/ad1012237

5. Page 11: “In our moderator analysis, we examined whether the effects of music listening on stress recovery were reliable across general (neuroendocrine, physiological, psychological) and specific outcome types.”

I am not sure whether the moderator analysis allows any claims about the reliability of the effects across outcome types. In theory, an effect could be highly reliable across many outcome types, while at the same time still being clearly stronger for some outcome types than for others (hence being moderated by them), right? Wouldn’t it be more correct to state that it was assessed to what extent the size of the effect on stress recovery depended on outcome type (or some equivalent formulation)? I am no expert on this issue, but I invite the authors to reconsider their wording.

Thank you for your comment. Indeed, we utilized the term ‘reliable’ when in fact our intention was to assess the extent to which the size of the effect of music listening on stress recovery would differ, for example across outcome types. We have thus adjusted our wording to the following (page 12):

“In our moderator analysis, we examined whether the effects of music listening on stress recovery differed across general (neuroendocrine, physiological, psychological) and specific outcome types.”

6. There is a type on page 15: Wisagreements � Disagreements

Thank you for pointing this out. The typo (page 15) has been fixed.

“Disagreements were resolved through face-to-face discussions, or through consultation with SG and KR when no consensus could be reached.”

7. As the authors rightly point out on page 6, stress recovery involves a process in which “changes that have occurred in response to a stressor revert to pre-stress baselines”. To quantify stress recovery, it therefore seems crucial to take individual pre-stress baseline levels into account.

To the reader, it does not readily become clear whether the effects derived from the studies included in the meta-analysis indeed reflect the extent to which stress levels “return to baseline”. From Table 4, the included studies seem to be a mix of 1) studies reporting differences in change scores with respect to pre-stress baseline levels and 2) studies reporting raw group differences in post-music stress levels. This may require some sort of disclaimer.

Could you please reflect on these analytical differences and their (possible) implications for the interpretation of your meta-analysis, in relation to the term “recovery”?

Previous research has shown that following an acute stress reaction, all elevated physiological and psychological parameters will naturally revert to pre-stress baselines within 30-60 minutes (Hermans et al., 2014). As such, the most immediate proof of the effect of music listening on stress recovery would be to see whether listening to music would allow participants to reach their respective baseline levels sooner within time frame. Unfortunately, as we also point out in our Discussion, it is rare for studies to adopt a design where such changes are monitored, particularly through use of continuous measures. Instead, as you have rightly pointed out, studies either compare post-stress and post-manipulation change scores between conditions or compare post-manipulation raw group differences between music and comparable control conditions. The effects of music listening on stress recovery that we describe in our meta-analysis thus reflect how reactive participants’ stress recovery processes are when listening to music, rather than how soon participants recover, with the assumption that greater reactivity (e.g., larger decreases in heart rate) post-stressor also results in earlier returns to baseline.

Hermans, E. J., Henckens, M. J., Joëls, M., & Fernández, G. (2014). Dynamic adaptation of large-scale brain networks in response to acute stressors. Trends in Neurosciences, 37, 304-314. https://doi.org/10.1016/j.tins.2014.03.006

8. On page 37-38 you write: “Khalfa et al. [55] reported that post-stressor cortisol decreased more rapidly for participants who listened to experimenter-selected classical music, compared with participants who sat in silence”

In Table 2 you write, when referring to this study: “Increase in post-stressor cortisol for music group significantly lower compared to control group (+)”.

These descriptions differ – could you please adapt the main text to match the (correct) description in the table?

Thank you for pointing this out. The correct description was what we wrote in text. We have thus adjusted the description in Table 4 to:

“Significant, rapid decrease in post-stressor cortisol in music group compared to control group (+).”

9. On page 40-41, you write: “While previous reviews suggest that music-based interventions may be moderately beneficial for stress-related outcomes, particularly in medical and therapeutic settings, our results suggest that the magnitude of this effect for healthy individuals may be more modest.”

While I largely agree with the contents of this paragraph (and with the further comments on this issue on page 46), I think the term “healthy individuals” (to label the category for which music is less effective for stress recovery) does not capture the essence of the differences between the different types of studies, and hence using this term may be a bit misleading.

As is stated further down the paragraph, the stress in studies conducted in medical and therapeutic settings likely has a more protracted time course, which does not directly have to do with the participants being (not) healthy. Furthermore, stress may differ in intensity between laboratory and medical real-life/settings, and the effectiveness of music may depend on the research setting as well.

It would be great if you could somewhat adapt the wording of this paragraph, to avoid the impression that the (non-) effectiveness of music depends on the participants being healthy. Rather, it seems more likely that several (interrelated) factors associated with the different research settings (e.g., type, intensity, and duration of stress) are driving these differences. You might e.g., use the term “healthy individuals under brief, experimentally induced stress” instead.

Thank you for your comment. Indeed, the term “healthy individuals” may at times oversimplify the fact that the effect of music listening may differ based on differences in research settings. We have reworded the paragraph as follows:

“The results of our review contrast those of previous meta-analyses, which underscore the relevance of music-based interventions for stress-reduction [11, 12]. While previous reviews suggest that music-based interventions may be moderately beneficial for stress-related outcomes, particularly in medical and therapeutic settings, our results suggest that the magnitude of this effect outside of these settings, particularly for healthy individuals under acute, experimentally induced stress, may be more modest. We presume that one of the principal reasons for this difference was our decision to exclude studies conducted in medical and therapeutic settings. In previous reviews, randomized controlled trials of the effects of music-based interventions within medical and therapeutic settings constituted a large portion of included studies: 67 of 79 (85%) studies in de Witte et al. [11], and 15 of 22 (68%) studies in Pelletier [12], making it more likely that overall effect sizes were derived from studies conducted within these settings. Tentatively, the effects of music listening may be more prominent for the stress recovery of individuals in medical or therapeutic contexts, compared to that of individuals under acute stress in an experimental context. Whereas the time course of stress responses and stress recovery in experimental settings can be considered relatively brief [25, 27, 41, 85], the time course of stress responses and stress recovery within medical and therapeutic settings may be significantly more protracted [13, 14]. Thus, within medical and therapeutic settings, music may be exerting its influence on neuroendocrine, physiological, and psychological processes that have been subjected to longer periods of strain [28, 108].”

---------------

 

Response to Reviewer #2

Dear Reviewer,

Thank you for taking the time to review our manuscript. We appreciate the kind words you have mentioned on the execution of our review. More importantly, we sincerely appreciate the critical comments you have provided on our search strategy and adequacy of our approach. Below, we address your concerns in a point-by-point fashion and summarize the changes we have made to our manuscript based on your suggestions. In the response letter we have included with our revision, you will see your comments in bold, our responses in regular text, and excerpts from our manuscript in italics (unless otherwise indicated):

Major Concerns:

1) The authors identified 14 studies that are quite heterogenous in nature. I ask myself whether the approach of a meta-analysis is adequate for this rather small number of studies given this vast heterogeneity. Furthermore, not all studies were successful in stress induction. Wouldn't it be reasonable to exclude these studies from the analysis?

We acknowledge that the relatively small number of heterogeneous studies may render the results of our meta-analysis less meaningful. We can thus understand if concerns are raised about whether a meta-analysis is the most appropriate approach to synthesize the available empirical evidence on the relationship between music listening and stress recovery.

As we state in the Limitations of our manuscript (page 49), we are aware that the small number of included studies makes it difficult to draw meaningful, substantial conclusions based on the results of the meta-analysis alone. For this reason, we have supplemented the quantitative synthesis of the meta-analysis with a more qualitative synthesis from a systematic review. We think this combined approach has yielded a more nuanced review, as the qualitative description of the included studies have helped provide more context to the results of our meta-analysis. We have reported the systematic review in our Results section on page 27.

Thus, we think our review is still valuable despite the small number of studies, as it provides not only a quantitative synthesis of available evidence, but also provides a qualitative description of the potential sources of heterogeneity that the meta-analysis could not account for.

Next, thank you for mentioning your concern over the inclusion of studies whose stress induction procedures were not successful. A similar point was raised by another reviewer. We thus agree that it is particularly important to further address the inclusion of studies with unsuccessful stress induction checks.

In the previous version of our manuscript, three studies were coded to have ‘unsuccessful’ stress induction checks. One of these studies (Scheufele, 2000) was erroneously coded, as the author did report a marked increase in heart rate following their stress induction procedure compared to baseline. Meanwhile, in the remaining two studies (Gan et al., 2016 & Labbé et al., 2007), the authors have hinted that their respective stress induction procedures were successful. However, we coded these as ‘unsuccessful’ since the statistical analyses comparing post-stressor and baseline values of their stress recovery outcomes were missing. We ultimately decided to still include these two studies in the meta-analysis given that the reported mean scores for certain stress recovery outcomes still suggested there was an increase from baseline from which participants could recover from. For example, in Gan et al. (2016), mean state anxiety for their three conditions during the stress task were msedative = 46.97, mstimulative = 50.51, and mcontrol = 52.00, compared to baseline means scores of msedative = 38.80, mstimulative = 38.09, and mcontrol = 36.17. 

Based on this, we have thus updated the results of our analysis for the “stress induction checks” moderator (page 26), to reflect the change in coding for Scheufele (2000):

“Stress induction checks. There were no significant differences in the effects of music listening on stress recovery for studies with successful (g = 0.17301, 95% CI [-0.2635, 0.6155], p = .399625) and unsuccessful (g = 0.062361, 95% CI [-0.0894, 0.201.66], p = .115355) stress induction checks, β1 = -0.108257, p = .525661.”

Next, we have added a paragraph to acknowledge the inclusion of studies with unsuccessful stress tasks in our Discussion (page 48):

“Two studies with less successful stress induction procedures were still included in the review, given that reported raw scores for certain recovery outcomes still suggested an increase from baseline that participants could recover from. For example, in Gan et al. (2016), mean state anxiety for their three conditions during the stress task were msedative = 46.97, mstimulative = 50.51, and mcontrol = 52.00, compared to baseline means scores of msedative = 38.80, mstimulative = 38.09, and mcontrol = 36.17. Given that the overall estimated effect of music listening on the recovery process of healthy individuals following laboratory stressors may be relatively modest, it becomes particularly important to ensure that a sufficient stress response is elicited, to provide a larger window of opportunity in which the effect of music listening may be exerted on participants’ recovery processes. We thus encourage future studies to adopt validated (variations of) well-known stress tasks, such as the TSST, SECPT, or CO2 stress task, which have been demonstrated to consistently elicit marked physiological and psychological stress-related responses in laboratory settings.”

Gan, S. K. E., Lim, K. M. J., & Haw, Y. X. (2016). The relaxation effects of stimulative and sedative music on mathematics anxiety: A perception to physiology model. Psychology of Music, 44(4), 730-741. https://doi.org/10.1177/0305735615590430

Labbé, E., Schmidt, N., Babin, J., & Pharr, M. (2007). Coping with stress: the effectiveness of different types of music. Applied Psychophysiology and Biofeedback, 32, 163-168. https://doi.org/10.1007/s10484-007-9043-9

Scheufele, P. M. (2000). Effects of progressive relaxation and classical music on measurements of attention, relaxation, and stress responses. Journal of Behavioral Medicine, 23, 207-228. https://doi.org/10.21236/ad1012237

2) Concerning the search strategy, I wondered that PubMed was not included. Searching this data base might be useful as the number of studies identified is quite small.

One of the goals of our review was to highlight the overall effect of music listening on stress recovery in healthy individuals. This meant excluding, for example, studies on the effects of music listening in the management of treatment anxiety or stress during pregnancy and labor. As we mention in the Introduction of our review (page 3), we reasoned that the nature of stressors in medical and therapeutic settings, along with their subsequent recovery processes, would be difficult to generalize to more daily settings.

From our experience, most studies on music listening published on the PubMed database reported experiments conducted within medical or therapeutic settings. Thus, when designing our search strategy, we made the decision to exclude the PubMed database from our search.

Despite this, based on your comment, we conducted an additional search in the PubMed database using the same search strategy listed in Appendix A. We limited the additional search to studies published until April 2021 to match our original search. This additional search returned 958 studies, but none of these studies met our inclusion criteria. Our search in PubMed thus resulted in no additional studies.

We have reported this additional search in our manuscript on page 10:

“The results of this primary search were supplemented with three additional electronic searches in the publication databases of Web of Science, PsycINFO, and PubMed. Appendix A provides a complete description of our search terms. Together, this first step resulted in 3124 articles.”

We have also updated Figure 1 to include the addition of the PubMed search:

Information in the paragraph following Figure 1 has also been updated to reflect the additional search:

“During this initial screening, 3008 articles were excluded. KA then scanned the reference lists of the 116 remaining articles for potentially relevant studies, resulting in an additional three articles. Together, this second step resulted in 119 full-text reports to be assessed for eligibility.”

3) Inclusion and exclusion criteria are not specifically justified. For example, in Figure 1 exclusion criteria are presented, e.g. 'no music presented after stressor'. This might be the reason why the study of Thoma et al. (2013) (https://pubmed.ncbi.nlm.nih.gov/23940541/) is not party of the review, although I consider it highly relevant in the context of music listening and stress reduction. Overall, I would describe and justify more in detail criteria for inclusion and exclusion of studies.

Thank you for pointing this out. We agree that our inclusion and exclusion criteria could be better justified. As such, we have described our inclusion and exclusion criteria (pp. 10-11) more extensively, as follows:

“Lastly, KA used the following criteria to assess full-text reports for eligibility:

First, to minimize between-study heterogeneity, and to ensure that included studies investigated the effects of music listening on stress recovery as precisely as possible, studies must employ an experimental design including stress induction, with random assignment of participants to experimental and control conditions. Quasi-experimental studies were included only when they incorporated a control or comparison group. Second, to ensure that included studies tested the immediate effect music listening may have on the stress recovery process, studies should compare music listening to silence or an auditory stimulus (e.g., white noise, audiobooks) following stress induction. Third, to demonstrate this effect, studies must include at least one measure of neuroendocrine (e.g., cortisol), physiological (e.g., heart rate, blood pressure), or psychological (e.g., subjective stress, positive and negative affect) stress recovery outcome. Fourth, given that stress reactivity and recovery responses differ between children and adults, and with consideration to the potential role of music in the prevention of stress-related diseases in adults, studies must include healthy, adult, human participants. Fifth, to improve the generalization of our results in the context of daily stress recovery, studies where stress recovery occurred within a medical or therapeutic context, such as a hospital or operating room, were excluded.”

We agree that the findings of Thoma et al. (2013) are interesting and particularly relevant in the context of music listening and stress reduction. The experiment by Thoma and colleagues convincingly demonstrated that listening to music prior to a stressor resulted in a milder stress response compared to silence, which in turn resulted in a lower need for subsequent recovery. Although their finding speaks to the benefits of music listening in attenuating the stress response, their finding did not completely fit the scope of our review, which was the immediate effect of music listening on recovery from stress.

4) Introduction: first paragraph: I really like the introduction to the topic, as the aspect of stress in daily life is emphasized. I wondered why the authors did not include ambulatory assessment/ecological momentary assessment studies in their review, as these studies have high ecological validity. I recommend to expand the review and to assess independently the evidence concerning controlled studies with high internal validity on the one hand and daily life studies with high ecological validity on the other hand.

When we planned our review, we reasoned that focusing on experimental studies with high internal validity would allow us to examine the strongest available evidence on the presumed relationship between music listening and stress recovery. Furthermore, we hoped that, by focusing on experimental studies, between-study heterogeneity would thus be somewhat minimal – this was eventually not the case.

We agree that there is much to be gleaned from specifically investigating EMA studies on music listening and stress recovery, including further insight into interindividual differences when listening to music for the purpose of stress recovery, and how stress recovery outcomes may be influenced by music listening over time. Despite this, the inclusion of EMA studies in our review would have made it more difficult to determine the immediate effect of music listening on recovery from stress. Given the relatively lower control in EMA studies (e.g., the absence of a clear control condition), claims about causality may be trickier to draw from EMA studies compared to experiments. Furthermore, given that measurements occur outside of the laboratory, it becomes difficult to rule out the effects of contextual variables, particularly when they are not explicitly accounted for in the design of an EMA study. As such, we respectfully argue against the inclusion of EMA studies in our current review, given the stronger ‘causal’ evidence that may be derived from experimental studies, and because we agree that evidence from experimental and EMA studies should be assessed independently of each other due to differences in contextual factors.

5) It may be my personal opinion, but I was irritated by the many instances the authors use the term 'beliefs', e.g., abstract 'given the popular and widespread belief'. My recommendation is to re-word this and acknowledge the empirical evidence underlying this statement.

We apologize for the discomfort we have caused you as you reviewed our manuscript. We agree that, in principle, it is good to acknowledge available empirical evidence rather than labeling a statement a ‘belief’. We have thus adjusted the following instances of the term ‘belief’, starting with the Abstract:

“Studies suggest that listening to music is beneficial for stress reduction. Thus, music listening stands to be a promising method to promote effective recovery from exposure to daily stressors.”

On page 7:

“Furthermore, studies have demonstrated that listening to music may influence mood [59, 60].”

Finally, in our Conclusion:

“Studies commonly suggest that listening to music may have a positive influence on stress recovery”

Minor Concerns:

Abstract: 'beneficial for stress' should be specified (beneficial for stress reduction)

Abstract: please report how many participants in total were included in the 14 studies

Abstract: please specify that (randomized)-controlled studies were included

We have added the above suggestions to the Abstract:

“Studies have suggested that listening to music may be beneficial for stress reduction. Thus, music listening stands to be a promising method to promote effective recovery from exposure to daily stressors.”

“As such, to clarify the current literature, we conducted a systematic review with meta-analysis of randomized, controlled experimental studies investigating the effects of music listening on stress recovery in healthy individuals.”

“In fourteen experimental studies, participants (N = 706) were first exposed to an acute laboratory stressor, following which they were either exposed to music or a control condition.”

Introduction: 'It is a popular and widespread opinion that music may be beneficial for

stress recovery [10]': I am not convinced that the citation is that adequate in this context. Levitin demonstrates in this book many instances for beneficial effects of music. I am not satisfied with labeling his statement as 'popular and widespread opinion'. Furthermore, I do not consider this book an optimal citation for this peer-reviewed journal article as there is a vast body of empirical evidence available.

Thank you for mentioning this. We have rewritten the sentence and provided an alternative reference for it. Below is an excerpt from the paragraph (page 3), with the new sentence highlighted in bold:

“Various activities have been proposed that may lead to better stress recovery, one among them being music listening. Music listening may have a modulatory effect on the human stress response (Thoma et al., 2013). Furthermore, given that music is readily available through online streaming services, music listening stands to be a time- and cost-effective method to facilitate daily stress recovery.”

Thoma, M. V., La Marca, R., Brönnimann, R., Finkel, L., Ehlert, U., & Nater, U. M. (2013). The effect of music on the human stress response. PLOS ONE, 8, e70156. https://doi.org/10.1371/journal.pone.0070156

Introduction l.60: My first thought was to question the necessity of this review given the fact that an extensive review was just published. Although, the authors justify their review in the ensuing paragraph, I would recommend to state immediately more clearly that the scope of the other review was different.

Thank you for your suggestion. We agree that the urgency of our review could be stated earlier in the manuscript. We have restructured the two paragraphs as follows:

“Furthermore, given that music is readily available through online streaming services, music listening stands to be a time- and cost-effective method to facilitate daily stress recovery. Indeed, a recent meta-analysis of 104 randomized controlled trials on the effects of music concluded that music-based interventions have a positive impact on both physiological (d = .380, 95% CI [0.30–0.47]) and psychological (d = .545, 95% CI [0.43–0.66]) stress-related outcomes [11]. However, a large proportion of studies included in this meta-analysis were conducted in medical or therapeutic settings, and the included music-based interventions encompassed not only music listening but also music therapy. Thus, a more specific review to determine whether music listening alone is beneficial for the recovery of healthy individuals outside medical and therapeutic settings seemed justified.”

Introduction l.78: Why not include studies demonstrating effects on cortisol?

We have added studies demonstrating equivocal effects on cortisol to the sentence in question (page 4):

“Indeed, studies on music listening and stress recovery in healthy individuals are equivocal: although music listening is considered beneficial for physiological stress recovery, several studies have reported no differences in heart rate, heart rate variability, respiration rate, or blood pressure, or cortisol recovery between participants who listened to music and those who either sat in silence or listened to an auditory control [18-21].”

l.133: I would not recommend to write 'smaller amounts of salivary alpha-amylase'. Rather, less activity of alpha-amylase.

We agree that ‘less activity of alpha-amylase’ is more appropriate given what the outcome represents. We have replaced ‘amounts of salivary alpha-amylase’ accordingly (page 6):

“This manifests as a restoration of parasympathetic activity, marked by a deceleration of heart rate and respiration rate, lower systolic and diastolic blood pressure, and less activity of salivary alpha-amylase [4, 29-32].”

l.153: Study 54 refers to an ambulatory assessment study – therefore, there was no control to silence or noise in a comparable manner to experimental studies. Please re-word.

Thank you for pointing this out. We have adjusted the sentences accordingly (page 7):

“For example, music listening has been associated with lower heart rate [49-51], systolic blood pressure [22, 50, 52], skin conductance [18, 20, 53, 54], and cortisol [55] compared to silence or an auditory control condition. Similarly, participants who listened to music following stress demonstrated less activity of salivary alpha-amylase and lower cortisol compared to when music was listened to for other purposes [56].”

l.164: I do not agree with the statement that the candidate mechanism underlying beneficial effects of music has already been identified. I would rather prefer to see here a more comprehensive statement acknowledging that the exact underlying mechanisms remain to be elucidated and that different notions exist, e.g., literature by Koelsch…

Thank you for your suggestion. It was not our intention to suggest that a definite mechanism underlying the beneficial effects of music listening on stress recovery has been identified. We agree that a more comprehensive statement would help convey this point more clearly. Following your suggestion, we have adjusted the paragraph accordingly (page 7-8):

“The exact mechanisms underlying the effects of music listening on stress recovery remain to be elucidated. Music-evoked positive emotions are thought to be particularly beneficial for stress recovery, as they may help undo the unfavourable changes wrought by negative emotions during stress, ultimately aiding the stress recovery process (Tugade & Fredrickson, 2004). Alternatively, music-evoked emotions may promote a more robust, and thus more adaptive, stress response (Koelsch et al., 2016), which may be followed by an equally robust period of recovery. Next, it has been theorized that music may act as an anchor that draws attention away from post-stressor ruminative thoughts or negative affective states, thus preventing a lengthening of physiological activation, and facilitating a more regular stress recovery process (Baltazar et al., 2019; Radstaak et al., 2014). Finally, physiological rhythms in our body, such as respiration, cardiovascular activity, and electroencephalographic activity, may become fully or partially synchronized with rhythmical elements perceived in music (Ellis & Thayer, 2010; Trost et al., 2017). This rhythmic entrainment process is thought to occur via a bottom-up process that originates in the brainstem: salient musical features, such as tempo, pitch, and loudness, are continuously tracked by the brainstem, generating similar changes in ANS activity over time…”

Baltazar, M., & Saarikallio, S. (2019). Strategies and mechanisms in musical affect self-regulation: A new model. Musicae Scientiae, 23(2), 177-195. https://doi.org/10.1177/1029864917715061

Ellis, R. J., & Thayer, J. F. (2010). Music and autonomic nervous system (dys) function. Music perception, 27(4), 317-326. https://doi.org/10.1525/mp.2010.27.4.317

Koelsch, S., Boehlig, A., Hohenadel, M., Nitsche, I., Bauer, K., & Sack, U. (2016). The impact of acute stress on hormones and cytokines and how their recovery is affected by music-evoked positive mood. Scientific reports, 6(1), 1-11. https://doi.org/10.1038/srep23008

Radstaak, M., Geurts, S. A., Brosschot, J. F., & Kompier, M. A. (2014). Music and psychophysiological recovery from stress. Psychosomatic medicine, 76(7), 529-537. doi: 10.1097/PSY.0000000000000094

Trost, W. J., Labbé, C., & Grandjean, D. (2017). Rhythmic entrainment as a musical affect induction mechanism. Neuropsychologia, 96, 96-110. https://doi.org/10.1016/j.neuropsychologia.2017.01.004

Tugade, M. M., Fredrickson, B. L., & Barrett, L. F. (2004). Psychological resilience and positive emotional granularity: Examining the benefits of positive emotions on coping and health. Journal of personality, 72(6), 1161-1190. https://doi.org/10.1111/j.1467-6494.2004.00294.x

l.274: IgA is named as outcome measure but has not been introduced. As it is an immune marker, the introduction should contain some information on interactions among stress and immune system.

Thank you for the reminder. We have added an additional sentence in the Introduction to present salivary IgA as a marker for stress (page 5):

“This process enables rapid, non-genomic effects that sustain ANS-mediated changes for the duration of the stressor, while suppressing immune system function [33-35]. This suppression is visible through lower concentrations of immunoglobulins, such as salivary immunoglobulin-A (s-IgA; Chojnowska et al., 2021).”

Chojnowska, S., Ptaszyńska-Sarosiek, I., Kępka, A., Knaś, M., & Waszkiewicz, N. (2021). Salivary biomarkers of stress, anxiety, and depression. Journal of Clinical Medicine, 10(3), 517. https://doi.org/10.3390/jcm10030517

l.310: some typos need revision

Thank you for pointing this out. The typos have been revised (page 15):

“Disagreements were resolved through face-to-face discussions, or through consultation with SG and KR when no consensus could be reached.”

l.425: Can you please indicate the range of music duration? As there is literature available on the effects of different music durations on beneficial effects, I assume that the range was very limited among these studies. Therefore, I would not state that duration is not important. Rather, that the range in experimental studies is not vast enough to allow for meaningful comparisons.

We have added the range of music duration in our report of the moderator analyses (page 26):

“Duration of music. There was no evidence that the effect of music listening on stress recovery may differ depending on how long participants were exposed to music, β1 = -0.005, p = .870 (rangeduration = 2 – 45 minutes).”

---

## [Decision Letter · Decision Letter 1]

5 Apr 2022

PONE-D-21-22806R1Music listening and stress recovery in healthy individuals: A systematic review with meta-analysis of experimental studiesPLOS ONE

Dear Dr. Adiasto,

Thank you for submitting your revised manuscript to PLOS ONE.  Both reviewers agree that your manuscript has greatly improved. One reviewer, however, has a few additional issues for you to consider. Therefore, we invite you to submit a revised version of the manuscript that addresses the points raised during the review process.

We look forward to receiving your revised manuscript.

Kind regards,

Urs M Nater

Academic Editor

PLOS ONE

Journal Requirements:

Reviewers' comments:

Reviewer's Responses to Questions

**Comments to the Author**

1. If the authors have adequately addressed your comments raised in a previous round of review and you feel that this manuscript is now acceptable for publication, you may indicate that here to bypass the “Comments to the Author” section, enter your conflict of interest statement in the “Confidential to Editor” section, and submit your "Accept" recommendation.

Reviewer #1: All comments have been addressed

Reviewer #2: (No Response)

2. Is the manuscript technically sound, and do the data support the conclusions?

Reviewer #1: Yes

Reviewer #2: Yes

3. Has the statistical analysis been performed appropriately and rigorously? 

Reviewer #1: Yes

Reviewer #2: Yes

4. Have the authors made all data underlying the findings in their manuscript fully available?

Reviewer #1: Yes

Reviewer #2: Yes

5. Is the manuscript presented in an intelligible fashion and written in standard English?

Reviewer #1: Yes

Reviewer #2: Yes

6. Review Comments to the Author

Reviewer #1: The authors have satisfactorily addressed all my comments. I am satisfied with the manuscript in its current format, and recommend it for publication.

Reviewer #2: The authors clearly put a lot of work and time into the revisions, I highly appreciate that, and I think that the manuscript is now much stronger; I only have a few concerns left to be addressed:

1) When reading your response letter I considered it an excellent idea to combine both systematic review and meta-analytic approach. However, when I read the manuscript, I felt it overloaded the paper. Furthermore, I did not really understand, why the number of included studies varies among these two approaches (14 vs. 17 studies).

My suggestion would be to either start with describing the review approach and then calculate the overall meta-analytic effect or to move the systematic review to the Appendix (after having adjusted the analysis to the same number of studies included).

2) Thank you for describing in more detail how you operationalized unsuccesful stress induction. I am not entirely convinced by your approach. For example, you provide mean statistics for two studies and conclude that the mean difference represents a successful stress induction. At least, I would expect a citation backing this up or a statistical test considering mean and standard deviation. I wondered if it was more approproate to distinguish successful from unsuccessful and (third category) not reported. As for now, I would still argue to include only those studies with successful stress reduction (expecially given the unequal ratio that limits comparisons anyways).

3) Thank you for describign in more detail your inclusion criteria. As they do not cover 'music should have been played after the stressor', I still argue that the Thoma Paper should be included. Therefore, please change the inclusion criteria accordingly or include the paper. If you adjust the inclusion criteria, I would recommend to refer to the Thoma Paper in the discussion as time of intervention (before, during or after stressor) might be an important modulator.

4) I am sorry to read that you decided against EMA studies as I consider it a huge strength to combine both experimental and EMA evidence. I am not convinced that these two approaches should be studied separately, as they complement each other in a meaningful way. Also, I believe that including EMA studies would shed light on the heterogeneity as you have multiple time points and multiple contextual factors being repetatedly assessed over time. Nevertheless I accept your choice here, but recommend to acknowlegde EMA studies in the discussion (or outlook). Particularly as you describe in the introduction that music is so easily available, studying the mechanisms in daily life seems to be timely.

5) Please omit the following sentence from the manuscript as I am afraid that it does not reflect the findings on alpha-amylase appropriately.

Similarly, participants who listened to music following stress demonstrated less activity

of salivary alpha-amylase and lower cortisol compared to when music was listened to for other

purposes [56].”

7. PLOS authors have the option to publish the peer review history of their article (what does this mean?). If published, this will include your full peer review and any attached files.

Reviewer #1: **Yes: **Jasminka Majdandžić

Reviewer #2: No

---

## [Author Response · Author response to Decision Letter 1]

12 May 2022

Response to Reviewer #1

Dear Dr. Majdandžić,

Thank you for taking the time to review our revised manuscript. Given your expertise in the effects of music listening on stress and wound healing, we are grateful that you have recommended the manuscript for publication in its current form.

Response to Reviewer #2

Dear Reviewer,

Thank you for taking the time to review our revised manuscript. We once again appreciate your kind words on our work and are happy to read that you consider the manuscript to be stronger in its current form. Below, we address the additional issues you have raised and summarize the changes we have consequently made to our manuscript in a point-by-point fashion. You will first see your comments, followed by our responses and excerpts from our manuscript where applicable:

------

1) When reading your response letter I considered it an excellent idea to combine both systematic review and meta-analytic approach. However, when I read the manuscript, I felt it overloaded the paper. Furthermore, I did not really understand, why the number of included studies varies among these two approaches (14 vs. 17 studies).

My suggestion would be to either start with describing the review approach and then calculate the overall meta-analytic effect or to move the systematic review to the Appendix (after having adjusted the analysis to the same number of studies included).

Thank you for your suggestion. In one of the earliest drafts of our review, we chose to present the systematic review section prior to the meta-analysis. Thus, the meta-analysis served to quantify the extensive qualitative evidence we presented in the systematic review. In line with this approach, we initially decided that studies which did not (or could not) provide the necessary means and standard deviations to estimate effect sizes (in our case, Hedge’s g) would be excluded from the meta-analysis. However, since means and standard deviations are less relevant to a qualitative review, we decided that studies without means and standard deviations could still be included in the systematic review section instead of being excluded completely, provided they met the rest of our inclusion criteria. As such, in the previous version of our manuscript, 17 studies were part of the systematic review section, while only 14 of those studies were part of the meta-analysis section.

In the previous version of our manuscript, we attempted to briefly explain this through the following information:

“Based on these criteria, the final sample for the systematic review portion of our review consisted of 17 studies. Finally, for studies to be included in the meta-analysis portion of our review, means and standard deviations of stress recovery outcomes following stressor cessation must be available. Corresponding authors were contacted when this information was not available. When authors did not or could not provide the required information (e.g., due to data no longer being accessible), the outcome was dropped from the meta-analysis. Thus, following attempts to obtain missing information, the final sample for the meta-analysis portion of our review consisted of 14 studies.”

In the current version of our manuscript, the meta-analysis section is currently presented first, with the systematic review section painting a more detailed picture about the methodological heterogeneity between included studies. We understand that an extensive qualitative portion which directly follows a straight-forward quantitative synthesis can feel somewhat overwhelming. Thus, we have decided to follow your suggestion and moved the bulk of the systematic review portion to the Appendix of our manuscript.

To accommodate this change. we have adjusted the paragraph in our inclusion criteria as follows (page 11):

“Finally, for the purpose of the meta-analysis, means and standard deviations of stress recovery outcomes following stressor cessation must be available. Corresponding authors were contacted when this information was not available. When authors did not or could not provide the required information (e.g., due to data no longer being accessible), outcomes were dropped from the meta-analysis. Following attempts to obtain missing information, the final sample for our review consisted of 14 studies.”

Furthermore, we have added the following paragraph after the results of our moderator analyses, to direct readers’ attention towards the systematic review in the Appendix (page 26):

“To further illustrate the methodological heterogeneity among experimental studies on the effect of music listening on stress recovery, we provide a more extensive, qualitative overview of the included studies in Appendix C. A summary of this overview is presented in Table 4.”

------

2) Thank you for describing in more detail how you operationalized unsuccessful stress induction. I am not entirely convinced by your approach. For example, you provide mean statistics for two studies and conclude that the mean difference represents a successful stress induction. At least, I would expect a citation backing this up or a statistical test considering mean and standard deviation. I wondered if it was more appropriate to distinguish successful from unsuccessful and (third category) not reported. As for now, I would still argue to include only those studies with successful stress reduction (especially given the unequal ratio that limits comparisons anyways).

Thank you for mentioning your concern. In the two studies we have labelled ‘unsuccessful’ with regards to their stress induction procedures, the authors do hint at the success of their stressors in their respective manuscripts. However, since this success was not explicitly reported (e.g., through a comparison between baseline and post-stressor outcomes), we ultimately decided to label these as ‘unsuccessful.’

For example, in Gan et al. (2015), the authors cite a statistically significant paired-samples t-test comparing pre-stressor and post-music math anxiety in their no-music control group (only) as evidence that their stress induction procedure was successful for all their conditions. Fortunately, Gan et al. (2015) have reported pre- and post-stressor (i.e., pre-music) means and standard deviations for all our outcomes of interest. Thus, we were able to conduct our own paired-samples t-tests using pooled standard deviations, assuming a correlation of 0.5 between pre- and post-stressor outcome measures (Estrada et al., 2018). Our own t-tests indeed demonstrate that there is a significant increase in stress from which participants can recover from.

We were not able to employ a similar method for Labbé et al. (2007), as the authors did not explicitly report pre- and post-stressor means and standard deviations for our outcomes of interest. However, Labbé et al. (2007) presented the results of several F-tests with significant effects of time (under stress/pre-music vs. post-music) for all their conditions. Though these tests are not as accurate as a baseline vs. post-stressor comparison to evaluate the effects of stress induction procedures, we considered it plausible that a stress reaction had indeed occurred.

With these considerations in mind, we agree that labeling both studies as ‘unsuccessful’, with regards to their stress induction procedures, may not be the most correct decision. Thus, we have decided to follow your suggestion and change their coding to ‘unreported’ instead – in the sense that what the studies reported were not an explicit test of their stress induction procedures.

Our final consideration to keep these two studies with unreported stress induction procedures in the meta-analysis is that the estimated cumulative effect size excluding the two studies (i.e., comprising studies with successful stress induction only), g = 0.173, 95% CI [-0.26, 0.61], p = .399 (page 25):

“Stress induction checks. There were no significant differences in the effects of music listening on stress recovery for studies with successful (g = 0.173, 95% CI [-0.26, 0.61], p = .399) and unreported (g = 0.062, 95% CI [-0.08, 0.20], p = .115) stress induction checks, β1 = -0.108, p = .661.”

…does not significantly differ in magnitude or significance with the overall cumulative effect size including the two studies (i.e., comprising studies with successful and unreported stress induction), g = 0.15, 95% CI [-0.21, 0.52], p = 0.374 (page 23):

“Based on a meta-regression with RVE, the estimated overall effect of music listening on stress recovery was g = 0.15, 95% CI [-0.21, 0.52], t(13) = 0.92, p = 0.374.”

With regards to this matter, we have updated our discussion on page 38:

“Two studies with unreported stress induction procedures were still included in the review [17,84], as reported means for certain recovery outcomes still suggested an increase from baseline that participants could recover from. For example, with the information reported in Gan et al. [84], assuming a correlation of 0.5 between baseline and post-stressor measures of state anxiety, we estimated that their stress induction procedure elicited a significant increase in state anxiety in their sedative music (t(34) = 5.87, p < .001, mdiff = 8.17, SDdiff = 8.24), stimulative music (t(34) = 8.21, p < .001, mdiff = 12.42, SDdiff = 8.95), and control (t(34) = 13.15, p < .001, mdiff = 15.83, SDdiff = 7.12) conditions. As the overall estimated effect of music listening on the recovery process of healthy individuals following laboratory stressors may be relatively modest, it becomes particularly important to ensure that a sufficient stress response is elicited, to provide a larger window of opportunity in which the effect of music listening may be exerted on participants’ recovery processes. We thus encourage future studies to adopt validated, (variations of) well-known stress tasks, such as the TSST [109], SECPT [110], or CO2 stress task [111], which have been demonstrated to consistently elicit marked physiological and psychological stress-related responses in laboratory settings. Furthermore, we remind future studies to candidly report the results of their stress induction procedures to facilitate subsequent meta-syntheses of the effects of music listening on stress recovery.”

Estrada, E., Ferrer, E., & Pardo, A. (2019). Statistics for evaluating pre-post change: Relation between change in the distribution center and change in the individual scores. Frontiers in psychology, 9, 2696.

Gan, S. K. E., Lim, K. M. J., & Haw, Y. X. (2016). The relaxation effects of stimulative and sedative music on mathematics anxiety: A perception to physiology model. Psychology of Music, 44(4), 730-741.

Labbé, E., Schmidt, N., Babin, J., & Pharr, M. (2007). Coping with stress: the effectiveness of different types of music. Applied psychophysiology and biofeedback, 32(3), 163-168.

------

3) Thank you for describing in more detail your inclusion criteria. As they do not cover 'music should have been played after the stressor', I still argue that the Thoma Paper should be included. Therefore, please change the inclusion criteria accordingly or include the paper. If you adjust the inclusion criteria, I would recommend to refer to the Thoma Paper in the discussion as time of intervention (before, during or after stressor) might be an important modulator.

Thank you for pointing this out. We have revised our inclusion criteria to make it clearer that we elected to focus on studies looking at the effects of music listening on stress recovery after a stressor (page 11):

“Second, studies should compare music listening to silence or an auditory stimulus (e.g., white noise, audiobooks). To ensure that included studies tested the immediate effect of music listening on stress recovery, exposure to music, silence, or auditory stimuli must occur after the stress induction procedure.”

Next, given our focus, we agree that the timing of the music intervention may be an important moderator of the effects of music listening on stress recovery. Thus, we have added the following paragraph to our Discussion, referring to several studies investigating the effects of music listening at different timings, including the Thoma et al. (2013) paper (page 38-39):

“As the current review focused on the effects of music listening after a stressor, studies where music was played before or during a stressor were omitted from our analyses. However, several studies suggest that the timing at which music is played (i.e., before, during, or after a stressor) may influence its effects on stress recovery. For example, in Burns et al. [48], participants who listened to classical music while anticipating a stressful task exhibited lower post-music heart rate compared to participants who anticipated the stressor in silence. Similarly, concentrations of salivary cortisol were lower for participants who watched a stressful visual stimulus while listening to music compared to those who watched the same stimulus without music [112]. Together, these findings hint that, when listened during a stressor, music may attenuate cortisol responses [9,113], thus reducing the subsequent need for recovery. On the other hand, Thoma et al. [9] reported that participants who listened to music prior to a stressor exhibited higher post-stressor cortisol compared to participants who listened to an audio control. Interestingly, despite the stronger stress response, Thoma et al. [9] noted a trend for quicker ANS recovery among participants who listened to music, particularly with regards to salivary alpha-amylase activity. This pattern of findings is consistent with the notion forwarded by Koelsch et al. [61], in that music listening may promote a more adaptive stress response, thus facilitating subsequent stress recovery processes. To date, research on timing differences in the context of music listening and stress recovery is scarce. Thus, future studies could further examine the influence of such timing differences to better understand their role in the relationship between music listening and stress recovery.”

------

4) I am sorry to read that you decided against EMA studies as I consider it a huge strength to combine both experimental and EMA evidence. I am not convinced that these two approaches should be studied separately, as they complement each other in a meaningful way. Also, I believe that including EMA studies would shed light on the heterogeneity as you have multiple time points and multiple contextual factors being repeatedly assessed over time. Nevertheless, I accept your choice here, but recommend to acknowledge EMA studies in the discussion (or outlook). Particularly as you describe in the introduction that music is so easily available, studying the mechanisms in daily life seems to be timely.

Thank you for your explanation, and we apologize to not have been able to account for EMA studies at the present time. We have followed your suggestion and acknowledged the value of EMA studies in our Discussion section (page 39):

“Given the pervasiveness of stress, Ecological Momentary Assessment (EMA) studies may provide a more intimate outlook on the dynamics of daily music listening behaviour, particularly for the purpose of stress recovery. For example, through an ambulatory assessment study, Linnemann et al. (38) revealed that music produced the most notable reductions in physiological and psychological stress outcomes when it was listened to for the purpose of ‘relaxation’, compared to other reasons such as ‘distraction’, ‘activation’, and ‘reducing boredom’. Indeed, given their high ecological validity, EMA studies may provide further insight into important contextual variables in the relationship between music listening and stress recovery. For example, in an EMA study, listening to music in the presence of others was related to decreased subjective stress, attenuated cortisol secretion, and higher activity of salivary alpha-amylase (55). Furthermore, physiological responses to music may co-vary between members of a dyad when music is listened to by couples (114). Thus, given the benefits of EMA studies, we invite future studies to continue exploring the dynamics and contextual factors of music listening behaviour for stress recovery in daily life.”

------

5) Please omit the following sentence from the manuscript as I am afraid that it does not reflect the findings on alpha-amylase appropriately.

Similarly, participants who listened to music following stress demonstrated less activity

of salivary alpha-amylase and lower cortisol compared to when music was listened to for other purposes [56].”

Our apologies for not appropriately conveying the effects of music listening on salivary alpha-amylase. The sentence in question has been deleted. The paragraph now reads: 

“For example, music listening has been associated with lower heart rate [48–50], systolic blood pressure [21,49,51], skin conductance [17,19,52,53], and cortisol [54,55] compared to silence or an auditory control condition. Furthermore, music listening has been associated with higher parasympathetic activity [56] compared to silence [3,37]. Together, these findings suggest that music listening may generate beneficial changes in ANS and HPA axis activity that should be conducive to the stress recovery process [27,57,58].”

---

## [Editor Report · Decision Letter 2]

3 Jun 2022

Music listening and stress recovery in healthy individuals: A systematic review with meta-analysis of experimental studies

PONE-D-21-22806R2

Dear Dr. Adiasto,

We’re pleased to inform you that your manuscript has been judged scientifically suitable for publication and will be formally accepted for publication once it meets all outstanding technical requirements.

Kind regards,

Urs M Nater

Academic Editor

PLOS ONE
---

## [Editor Report · Acceptance letter]

9 Jun 2022

PONE-D-21-22806R2 

Music listening and stress recovery in healthy individuals:
a systematic review with meta-analysis of experimental studies 

Dear Dr. Adiasto:

I'm pleased to inform you that your manuscript has been deemed suitable for publication in PLOS ONE. Congratulations! Your manuscript is now with our production department. 

Kind regards, 

on behalf of

Dr. Urs M Nater 

Academic Editor

PLOS ONE